# A Comprehensive Observational Based Multiphase Chemical Model Analysis of the Sulfur Dioxide Oxidations in both Summer and Winter

*Huan Song[1], Keding Lu[1]\*, Can Ye[1], Huabin Dong[1], Shule Li[1], Shiyi Chen[1], Zhijun Wu[1], Mei Zheng[1], Limin Zeng[1], Min Hu[1] & Yuanhang Zhang[1]*

State Key Joint Laboratory or Environmental Simulation and Pollution Control, College of Environmental

Sciences and Engineering, Peking University, Beijing, China

*Correspondence to: Keding Lu; ORCID ID: 12465

## Abstract

Sulfate is one of the main components of the haze fine particles and the formation mechanism remains controversial. Lacking of detailed and comprehensive field data hinders the accurate evaluation of relative roles of prevailing sulfate formation pathways. Here, we analysed the sulfate production rates using a state-of-art multiphase model constrained to the observed concentrations of transition metal, nitrogen dioxide, ozone, hydrogen peroxide, and other important parameters in winter and summer in the North China Plain. Our results showed that aqueous TMI-catalysed oxidation was the most important pathway followed by the surface oxidation of Mn in both winter and summer, while the hydroxyl and criegee radicals oxidations contribute significantly in summer. In addition, we also modelled the published cases for the fog and cloud conditions. It is found that nitrogen dioxide oxidation is the dominant pathway for the fog in a higher pH range while hydroperoxide and ozone oxidations dominated for the cloud.

## Introduction

Secondary sulfate aerosol is an important component of fine particles in severe haze periods (Zheng et al., 2015; Huang et al., 2014b; Guo et al., 2014), which adversely affect the environmental quality and human health (Lippmann and Thurston, 1996; Fang et al., 2017; Shang et al., 2020). Traditional atmospheric models evaluate secondary sulfate formation via the gas-phase oxidation of sulfur dioxide ($SO_2$) and a series of multiphase oxidation of dissolved $SO_2$ in cloud water. During haze events, multiphase oxidation of dissolved $SO_2$ is more important than $SO_2$ directly oxidized by gas-phase radicals (Atkinson et al., 2004; Barth et al., 2002) because of the significantly reduced ultraviolet (UV) radiation intensity due to the aerosol dimming effect. Gas-phase reactions, especially those favouring multiphase chemistry, cannot capture the high concentrations of sulfate aerosols during haze events. Moreover, rapid sulfate production is observed during cloud-free conditions indicating that aerosol multiphase oxidation may be important during haze periods (Moch et al., 2018). These effects cause a major gap between the measured sulfate concentrations under weak UV radiation and the concentrations calculated using traditional atmospheric

models.
Assessing the mechanism of multiphase secondary sulfate formation during haze periods helps evaluate
the effect of multiphase oxidation. While the gas-phase oxidation rate of $SO_2$ and OH is well constrained,
there are many uncertainties in the quantification of the relative contribution of each multiphase $SO_2$
oxidation pathway during haze periods. Multiphase oxidation pathways of dissolved $SO_2$ (Seinfeld and
Pandis, 2016; Liu et al., 2020a; Zhu et al., 2020a; Seigneur and Saxena, 1988; Li et al., 2020b) include
oxidation by (1) hydrogen peroxide ($H_2O_2$); (2) ozone ($O_3$); (3) transition metal ions [TMI, i.e., Fe (III)
and Mn (II)] catalysed oxidation pathway (aqTMI)]; and (4) Mn-catalysed oxidation of $SO_2$ on aerosol
surfaces pathway (Mn-surface) (Wang et al., 2021). Some studies (Cheng et al., 2016; Wang et al., 2016;
Xue et al., 2016; Li et al., 2018) also suggested that nitrogen oxides may play a crucial role in the
explosive growth of sulfate formation during severe haze days in Beijing because of the high pH near a
neutral system, by facilitating the catalysis of mineral dust (Liu et al., 2012; Zhao et al., 2018) or the
photolysis of nitrous acid (Zheng et al., 2020). However, the average pH during Beijing haze periods is
approximately 4.2 (Liu et al., 2017), and a high level of $NH_3$ does not increase the aerosol pH sufficiently
to yield $NO_2$-dominated sulfate formation (Guo et al., 2017). Other studies (Ye et al., 2018; Liu et al.,
2020b) emphasized the importance of $H_2O_2$ oxidation to sulfate formation due to the underestimation of
$H_2O_2$ concentrations during haze episodes in previous studies or the influence of high ionic strength ($I_s$) of
aerosol solutions on the $H_2O_2$ oxidation rate, which implies that oxidant concentrations for $SO_2$ oxidation
constrained to the observed values from field measurements are required. Previous study (Wang et al.,
2020) showed that photosensitization is a new pathway for atmospheric sulfate formation and requires
further verification. According to previous studies of the GEOS-Chem model and including the
measurements of oxygen isotopes ($\Delta^{17}O$ ($SO_4^{2-}$)) (He et al., 2018; Shao et al., 2019; Li et al., 2020a; Yue
et al., 2020), several studies showed that aqTMI was important during some haze periods. Overall, the
formation mechanisms of the missing sulfate sources remain unclear and controversial.
Sulfate formation is a complex multiphase physicochemical reaction process, in which parameters have
multiple interrelationships. The previous studies have mostly selected typical conditions with fixed
parameters for numerical calculations, ignoring the fact that sulfate formation is a complex dynamic
process. A comprehensive and explicit evaluation of the sulfate generation process requires real-time
feedback and explicit constraints of observational data. Therefore, it is crucial to apply constrained
parameters from field campaigns in the calculations. Moreover, as proposed in previous studies (Liu et
al., 2020b; Cheng et al., 2016), due to the lower water content in aerosol particles than in cloud water, the
non-ideality effects of aerosol solutions should be carefully considered.
In this study, we modelled the concentrations of the main reagents of sulphate formation reactions using a
state-of-art Peking-University-Multiple-phAse Reaction Kinetic Model (PKU-MARK) based on the data
measured in two field campaigns conducted in the winter and summer in the North China Plain (NCP)
where several particle pollutions happened. The non-ideality of aerosol solutions was considered in the
calculation of both gas solubility and aqueous-phase reaction rates. Chemical regimes in the aerosol
particle bulk phase were analysed to understand the role of gas-phase radical precursors, particle TMIs,
aerosol surface concentrations and the aerosol liquid water content (ALWC) on the aqueous reactant
levels and the sulfate formation rate. All particle concentrations reported are fine particle matters
particulate matter with aerodynamic diameter of 2.5 µm or less ($PM_{2.5}$).
The overall goal of this work is to evaluate the contribution of different secondary sulfate formation
pathways under actual field measurement conditions in the NCP. Effects of non-ideality of condensed
particle phase and the solubility of gas-phase reactants on the reactions enable the comparisons with
parameters previously obtained in model calculations. In addition, episodes at different pollution levels in
the winter and summer campaigns were selected to evaluate the contribution of prevailing sulfate
formation pathways proposed in previous studies. As a study evaluating the contribution of different
sulfate formation pathways during field campaign observations, this work provides an improved
understanding of atmospheric sulfate formation at different pollution levels in the NCP.


## 2 Results

### 2.1 Overview of the field observations

Table 1 shows the key meteorological parameters, trace gases concentrations, calculated ALWC, ionic

strength, pH and sulfate formation rates under different pollution conditions in PKU-17 and WD-14

comprehensive field campaigns. Sampling location and experimental methods used in these two

campaigns are summarized in the Method part. The pollution degree is classified according to the mass

concentration of $PM_{2.5}$. The clean condition means $PM_{2.5}$ smaller than 35 $\mu g/m^3$, the slightly polluted

condition is 35-75 $\mu g/m^3$, the polluted condition is 75-150 $\mu g/m^3$ and highly polluted is larger than 150

$\mu g/m^3$. Sulfate formation rates were modelled by the Multiple-phAse Reaction Kinetic Model (PKU-

MARK) (mentioned in Method) with constrained parameters. The effects of aerosol non-ideality were

considered in the size-segregated model. Data points with relatively humidity (RH) smaller than 20% and

AWLC smaller than 1 $\mu g/m^3$ were abandoned to improve the accuracy of the results.

Transition metals concentrations including Fe and Mn increased with PM mass (as shown in Fig.1).

Photochemical oxidants including $H_2O_2$ and $O_3$ exhibited a decreasing trend with the increase of PM mass

because of the significantly reduced solar ultraviolet (UV) radiation intensity due to the aerosol dimming

effect. Some studies reported high $H_2O_2$ concentrations during haze episodes (Ye et al., 2018), whereas in

PKU-17 field campaign, the average concentration of $H_2O_2$ was only 20.9±22.8 pptV in highly polluted

conditions. Higher sulfate concentration was observed in the high range of RH and ALWC indicating

their enhancement effects on the sulfate formation. We also picked four haze periods in PKU-17

observation, the time series of these key parameters are provided in Supplementary Information **(SI) Fig.**

**S4**.

Aerosol pH values were calculated using the ISORROPIA-II model. The calculated particle pH values as

shown in the **Table 1** are in good agreement with the values reported in other studies (Guo et al., 2017;

Weber et al., 2016). The lower pH in the range of 4.0–5.5 is beneficial to sulfate formation via aqTMI.

Aerosol liquid water is another key component, higher loading of aerosol liquid water is more conducive

to the occurrence of multiphase reactions. The ALWC in the PKU-17 and WD-14 campaigns was
calculated via the ISORROPIA II model with input concentrations of aerosol inorganic components (see
**Method M.3**). Aerosol liquid water did not freeze at winter temperatures below 273 K in the PKU field
campaign because of the salt induced freezing point depression (Koop et al., 2000). Wind speeds during
these haze events were persistently low (0.3–1.5 m/s), indicating the minor contribution of regional
transport to sulfate production.
Aqueous TMI concentration level is crucial in the evaluation of secondary sulfate formation in real
atmospheric conditions. Atmospheric anthropogenic sources of transition metals such as iron (Fe) are
crust related and the peak concentration of Fe in Beijing is correlated to the vehicle driving in traffic rush
hours. Copper (Cu), and manganese (Mn) are mainly from non-exhaust emissions of vehicles, fossil fuel
combustion or metallurgy (Alexander et al., 2009; Duan et al., 2012; Zhao et al., 2021). Concentrations of
transition metals are highly variable, ranging from <0.1 ng m-3 to >1000 ng m-3 globally (Alexander et
al., 2009). Fe solubility in atmospheric aerosols has been reported to range from 0.1% to 80% (Ito et al.,
2019; Hsu et al., 2010; Heal et al., 2005; Shi et al., 2012; Mahowald et al., 2005), and elevated levels of
Fe solubility have been observed in aerosols dominated by combustion sources. The average fractional Fe
solubility in areas away from dust source regions is typically between 5 and 25% (Baker and Jickells,
2006; Baker et al., 2006; Hsu et al., 2010). A recent study reported the average Fe solubility as 2.7–5.0%
in Chinese cities, and more than 65% of nano-sized Fe-containing particles were internally mixed with
sulfates and nitrates (Zhu et al., 2020b). The solubility of Mn tends to be higher than that of Fe (Baker et
al., 2006), which is 22–57% in urban aerosol particles (Huang et al., 2014c). In this study, we chose the
solubility of total Fe as 5% and total Mn solubility as 50% assuming that aerosol particles are internally
mixed. In Beijing, high concentrations of Fe, Cu, and Mn were observed (**Table S9**). Concentrations of
transition metals are strongly correlated during these haze periods; thus, we propose a fixed ratio of
Fe/Mn to account for the lack of Mn data in PKU-17 and WD-14 field campaigns (**SI Text S2**).
Aerosol trace metal speciation and water solubility are affected by factors such as photochemistry, aerosol
pH, and aerosol particle size (Baker and Jickells, 2006; Oakes et al., 2010). Soluble iron in aerosol water
exists as Fe (II) and Fe (III), with a series of redox recycling between the two species and other ions.
Partitioning between Fe (II) and Fe (III) varies diurnally with the highest fraction of Fe (II) found during
the day because of the photochemical reactions reducing Fe (III) to Fe (II). Photolysis reactions of iron
hydroxides and organic complexes were documented as the most important source of Fe (II) in cloud and
fog water. Oxalic acid and its deprotonated form, oxalate, have strong coordination ability with Fe and
form Fe-oxalate complexes, which have higher photochemical activity than Fe hydroxide. All these
mechanisms are included in the PKU-MARK model. Diurnal trends of sulfate formation were observed
during haze periods indicating the diurnal distribution of different states of iron. Redox cycling of other
TMIs such as Cu and Mn are also considered in the PKU-MARK model. Averaged percentage of soluble
Fe (III) and Mn (II) was 0.79% and 19.83% in winter polluted conditions and 2.57% and 52.15% in
summer polluted conditions. The main reason for the difference between winter and summer metal
solubility is that summer aerosols have higher water content and lower ionic strength, which is conducive
to the dissolution of Fe and Mn. The solubility range is in good agreement with the values reported in
previous observations (Ito et al., 2019; Hsu et al., 2010).
The influence of aerosol ionic strength on aqTMI reaction rates was considered carefully in the study.
Higher ALWC is typically accompanied by lower ionic strength, which increases the activity of TMI. The
relationship (Liu et al., 2020b) between the rate coefficients of the TMI pathway and ionic strength is
displayed in **Fig. S1**. The sulfate formation rate decreased by 424.82 times when ionic strength was 45 M
compared to the dilute solution with ionic strength of 0 M. Despite considering the effect of the activity
coefficient on the reaction rate of aqTMI, the contribution of the aqTMI was still dominant during haze
periods indicating that the dominance of aqTMI can be a widespread phenomenon, as recommended in
previous studies (He et al., 2018; Shao et al., 2019; Li et al., 2020a; Yue et al., 2020).
**2.2 Analysis of sulfate formation rate in different pollution conditions**
**Fig. 1 (a) and (b)** display the 3-h averaged sulfate formation rates in the PKU-17 and WD-14 during haze
periods. Contributions of the gas-phase radical oxidants were much higher during summer time. To fully
explain the relative contributions to sulfate formation from different pathways, the stabilized criegee
intermediates (SCIs) oxidant was also considered in the calculations. Based on the previous report
(Sarwar, 2013), the inclusion of the SCIs oxidation pathway further enhances sulfate production. We
modified the Regional Atmospheric Chemistry Mechanism (RACM2) (Goliff et al., 2013; Goliff and
Stockwell, 2008) to represent three explicit SCIs and their subsequent reactions (Welz et al., 2012) with
$SO_2$, $NO_2$, aldehydes, ketones, water monomer, and water dimmer and calculated the contribution of this
pathway in two field campaigns.
The contribution of aqTMI increased rapidly with the aggravated pollution. High concentrations of
transition metals observed in Beijing facilitated the dissolution of Fe, Cu, and Mn. The relationship of
ionic strength and aqTMI rate constant is illustrated in **SI Fig. S1** and **Table S2** (Liu et al., 2020b). αFe
(III) is defined as the product of the Fe (III) activity coefficient, concentration, molecular weight (56) and
aerosol liquid water content. Compared to the total Fe concentration, it is more effective to evaluate the
impact of αFe (III) on sulfate formation. The relationship between αFe (III) and SOR
($\equiv n(SO_2)/n(SO_2+SO_4^{2-})$, defined as the ratio of mole concentration of $SO_2$ with the sum of $SO_2$ and $SO_4^{2-}$
mole concentrations) in PKU-17 winter field campaign was shown in **SI Figure S5**. Because of the
inhibition of the effects of high ionic strength on the rate constant of aqTMI, a high volume of aerosol
water during the haze event increased the TMI activity coefficient benefiting sulfate formation. Obvious
correlations between αFe (III) and sulfate concentration shown in **Fig. 1 (c) and (d)** were observed in the
haze periods both in summer ($R^2$=0.63) and winter ($R^2$=0.71) and the correlation is consistent with the
important contributions from aqTMI pathway to the sulfate formation. Affected by the higher boundary
layer height and higher gas phase radical concentration in summer, the correlation between sulfate
oxidant ratio SOR and PM mass in summer is not as significant as that in winter. In summer, as illustrated
in **Fig. S6**, there was still an obvious positive dependence between SOR and RH and ALWC, whereas a
negative correlation was found between SOR and odd-oxygen ($[Ox]\equiv[O_3]+[NO_2]$). As shown in **Fig. 1 (e)**
**and (f)**, the sulfate formation through gaseous reaction was more important in summer than in winter,
mainly provided by gas phase radicals (OH and SCIs). In WD-14 field campaign, heterogeneous aqTMI
pathways were still dominant in the secondary sulfate formation.

## 2.3 Dependence of the Secondary sulfate formation rates on aerosol pH and water content


Aerosol pH and ALWC were calculated using the ISORROPIA-II model (Method M3). Because of the
high sensitivity of sulfate formation to pH, the lower range of aerosol pH during these two campaigns
made the aqTMI the most important one. The effects of high aerosol ionic strength on the dissolution
equilibrium and reaction rates were considered in calculations (Liu et al., 2020b) (**SI Table S2 to S4**).
Due to the low $H_2O_2$ concentration (~0.023 ppbV) and low ALWC observed in the PKU-17 field
campaign, the average contribution of $H_2O_2$ in haze periods ($PM_{2.5} > 75$ μg/m$^3$) was about $0.11\pm0.15$
μg/m$^3$/h. Higher gas-phase $H_2O_2$ concentration may further increase the contribution of this pathway to
sulfate formation. Based on a recent report (Ye et al., 2018), higher gas-phase $H_2O_2$ concentrations were
observed in the NCP during different haze events, including severe haze episodes in suburban areas. At
0.1 ppbV $H_2O_2$ (about five times higher than the observed $H_2O_2$ concentration), the calculated sulfate
formation rate was $0.52\pm0.76$ μg/m$^3$/h in haze periods with great uncertainty and still lower than the
contribution of the TMI pathway ($1.17\pm1.48$ μg/m$^3$/h).
Due to the potential interaction between various factors in the atmosphere, fixing certain parameters and
changing only the pH to obtain the sulfate production rate may cause errors. With the development of
haze, concentrations of $O_3$ and OH radicals decrease due to reduced UV radiation caused by the aerosol
dimming effect. Despite its minor contribution to sulfate production in winter, the increase in the ozone
oxidation rate with pH was slower under actual conditions. Contributions of gas-phase radicals also
showed a weak downward trend in the summer campaign (**Fig. 2 (c)**). The bias between calculated and
observed values indicated a dynamic balance of atmospheric oxidation in the gas phase and aerosol phase.
If we arbitrarily use the average values during haze periods and only changed the pH of the aerosols as in
previous studies, the obtained sulfate production rate will deviate from the observed values. Actual
ambient sulfate formation rates calculated using the measured values in polluted periods in two field
campaigns are illustrated in **Fig. 2 (a) and (c)**. Average values except for pH during the haze periods were
used to calculate the sulfate formation rates as shown in **Fig. 2 (b) and (d)**. The peak of the $H_2O_2$ line in
the figure is caused by the change in the water content and ionic strength. In the pH range of 4.0–6.0, the
calculated ALWC was in the highest range, increasing the contribution of $H_2O_2$ proportionally as
calculated using **equation (1)**.
Aerosol water content is another key factor that influences the contribution of different pathway to sulfate
formation. In the calculation, we changed the unit of sulfate formation rate from $\mu g/m^3_{air}$ to $mol/s \cdot L_{water}$
and the sulfate formation rate can be calculated via the following equation with the modeled
$\frac{dS(VI)}{dt}$ $(M/s)$:

223        $$\frac{dS(VI)}{dt} (\mu g \ m^{-3} \cdot h^{-1}) = 0.01 \times 3600 \ (s \ h^{-1}) \cdot 96 \ g \ mol^{-1} \cdot \frac{dS(VI)}{dt} (M \ s^{-1}) \cdot \frac{ALWC}{\rho_{water}} \qquad (1)$$

where ALWC is in units of $\mu g \ m^{-3}$ and $\rho_{water}$ is the water density in $kg \ L^{-1}$. At high ionic strength, this
expression is more accurate than the equivalent expression with the unit of $M \ s^{-1}$. The equilibrium amount
of $H_2O_2$, $O_3$, and $NO_2$ in units of $\mu g/m^3_{air}$ is controlled by the amount of ALW, ie there is equilibrium
between gas and particle water for these oxidants formed in the gas phase. And total amount of metal
elements, Fe, Cu or Mn is not dependent on aerosol water content. Aerosol water content does not affect
TMI levels in solution by affecting the solubility of the overall metal form of the specific species (**Fig.3**
shows insensitivity of pH to ALWC, which has been pointed out in other papers (Wong et al., 2020). The
reaction kinetics and rate constants summarized in **Table S2** suggest that there is a proportional
relationship between ALWC and sulfate formation pathways except aqTMI. One reason for the lower
sulfate formation rate observed in the PKU-17 $(1–3 \ \mu g \ m^{-3} \ h^{-1})$ is that the ALWC values were lower than
those assumed in previous studies $(ALWC = 300 \ \mu g \ m^{-3})$. This deviation from the ALWC significantly
reduces the contribution of several other pathways, but not the contribution of transition metals to sulfate
formation.
Due to the obvious heterogeneous reaction's contribution to sulfate formation in winter, we evaluated the
influence of ALWC on sulfate formation pathways in winter. TMI relevant pathways including aqTMI
and Mn-surface pathway were dominate in all range of ALWC as illustrated in **Fig.3**. In PKU-17 field
campaign, with the increasing of ALWC from 1 to 150 $\mu g/m^3$, the ratio of Mn-surface/aqTMI
continuously decreased mainly because of the decreasing particle specific surface areas. Mn-surface
contributed most in lower ALWC range where particle specific surface area was high and provide more
reaction positions. Aqueous transition metal ions mole concentration decreasing with the aerosol
hygroscopic growth indicating a "dilution effect" as shown in **Fig. S7** with the aerosol hygroscopic
growth, the increasing of transition metal total mass in air is slower than water mass in PKU-17. The ratio
of Fe total mass ($Fe_t$)/ALWC decreasing with $PM_{2.5}$ mass. Previous global scale observations (Sholkovitz
et al., 2012) of ~1100 samples also showed the hyperbolic trends of Fe solubility with total Fe mass.
Higher activity coefficients and lower aqueous TMI concentration led to the emergence of "high
platform" of the aqTMI pathways contribution to sulfate formation in the range of 50-150 $\mu g/m^3$ ALWC
(ie, higher effective aqueous TMI in this range). While ALWC exceeding 150 $\mu g/m^3$ in winter, the
increase of activity coefficients could not promote the rate of aqTMI. Due to the slight increase of aerosol
pH and the dilution effect of aerosol hygroscopic growth on TMI when ALWC exceeding 150 $\mu g/m^3$ as
discussed above, the importance of aqTMI and Mn-surface contributions were lowered. At this time, the
contributions of external oxidizing substances pathways such as $H_2O_2$, $NO_2$ or $O_3$ may rise in the proper
pH range as illustrated in **Fig.4**. In winter fog or cloud conditions with higher water content, the
contribution from TMI may decrease a lot for their low solubility and concentrations.
The same analysis also used in the summer WD-14 field campaign (as shown in **SI Fig.S8**). "The dilution
effect" occurred more dramatically in summer compared to that in winter because of a higher RH and
higher percentage of water in the aerosol. In this situation, the contribution of aqTMI or Mn-surface was
inhibited due to the low soluble TMI concentrations. Considering the positive relationships of SOR and
RH in summer WD-14 field campaign, aqueous and surface sulfate formation contributions mentioned in
the study could not explain the missing source of secondary sulfate. Because of the low pH range
observed in WD-14 field campaign, the contributions from $H_2O_2$, $NO_2$, $O_3$ or $NO_3^-$ photolysis were
negligible. The missing contribution may mainly come from other pathways such as photosensitizing
molecules (Wang et al., 2020) under stronger UV in summer or contributions from hydroxy methane
sulfonate (Moch et al., 2018; Ma et al., 2020) which need further studies.

# Discussion and Conclusion

We evaluated the contribution of different pathways to secondary sulfate formation using a state-of-art size-segregated multiphase model constrained to the observed parameters from two field campaigns in the North China Plain. In addition, the effects of aerosol solution non-ideality on aqueous-phase reaction rates as well as dissolution equilibriums were considered in the calculations. The results indicated that the aqueous TMI-catalysed oxidation pathway (aqTMI) was an important contributor to sulfate formation during haze episodes, which is consistent with the results of the isotope and WRF-CHEM studies (He et al., 2018; Shao et al., 2019; Li et al., 2020a; Yue et al., 2020).

Despite the dominant role of aqTMI in PKU-17 field campaign, contributions from other multiphase pathways are not negligible. Dominant pathways varied with conditions such as clear or haze periods in clouds or aerosol water. **Fig. 4** exhibits the contribution of different oxidation pathways to sulfate formation in aerosol water (under different pollution levels), fog, and clouds to indicate the dominant factors of sulfate formation under different conditions. In clear periods, gas-phase oxidation of $SO_2$ by gas phase radicals (OH and SCIs) happens continuously, contributing 0.01–0.6 μg/m$^3$/h to sulfate formation. At the clean time, sulfate production is mainly limited by relatively low $SO_2$ concentrations and low ALWC, which has promotion effects on the multiphase sulfate formation pathways. The average sulfate formation rate during clear days was 1.30 μg/m$^3$/h in winter and 2.13 μg/m$^3$/h in summer because of the generally higher ALWC in summer aerosol and much higher gas phase radical concentrations. Gas-phase radicals (OH and SCIs) continuously oxidize $SO_2$ during the haze and clear periods.

External oxidizing substances such as $NO_2$ and $O_3$ had little contribution to sulfate formation during these haze periods because of the high aerosol acidity. High pH (near 7) values were observed in these field campaigns when the contribution of the $NO_2$ pathway was dominant at some point but not during the entire pollution process; its proportion was much lower than that of aqTMI. Although the enhancement factor of $H_2O_2$ oxidation was considered based on the measurement of previous study (Liu et al., 2020b),

the contribution of $H_2O_2$ oxidation was still below 0.5 μg S(VI)/m$^3$/h because ALWC was about 10 times
lower than 300 μg m$^{-3}$, which was used in previous studies (Cheng et al., 2016; Liu et al., 2020b).
The sulfate formation rate is limited by the ALWC according to **equation (1)**. Aerosol particles have
lower water content than cloud droplets, which provides larger space for aqueous phase reactions.
Therefore, at the gas-phase $SO_2$ concentrations of 5–50 ppb, 10–100 times higher water content in fog and
cloud droplets can cause higher sulfate formation rates up to 100 μg m$^{-3}$ h$^{-1}$ assuming 0.1 g m$^{-3}$ water in
clouds (**Fig. 4)**. A high $H_2O_2$ concentration (1 ppb), which was 50 times higher than that in the PKU field
campaign, was used in the calculation in the Cloud_5.0 regime (Seinfeld and Pandis, 2016). No obvious
contribution from the $NO_2$ oxidation pathway was observed in the PKU-17 and WD-14 field campaigns
because of the lower pH range. As proposed in a previous study, the particulate nitrate photolysis can
explain the missing source of sulfate in Beijing haze (Zheng et al., 2020). However, according to the
recent laboratory report (Shi et al., 2021), the nitrate photolysis enhancement factor is no larger than 2 at
all RH ranges. We also included the calculation of nitrate photolysis in this study due to the high loading
of particle nitrate and found that the contribution was rather small (~0.008 μg m$^{-3}$ h$^{-1}$ in winter haze
periods); thus, we did not include this pathway in the figures.
According to our modelled results and the newest study (Wang et al., 2021), Mn surface reactions
contributed a lot to sulfate formation. Except for possible Mn(OH)$_x^{(3-x)}$ reacting with $SO_2$, Zhang et al.
(2006) proposed that other metal oxides such as $Fe_2O_3$ and $Al_2O_3$ can also react with $SO_2$ on the surface
of particles. While as mentioned above, the ratio of contributions from Mn-surface/ aqTMI to produce
sulfate will decrease with aerosol hygroscopic growth owning higher ALWC and lower specific surface
areas (as shown in **Fig.3** panel (b) black dotted line). What's more, the organic coating of aerosol particles
can largely reduce the reactivity of surface heterogeneous reactions (Zelenov et al., 2017; Anttila et al.,
2006; Folkers et al., 2003; Ryder et al., 2015) and may cause the Mn-surface pathway less important.
High mass concentrations of organic aerosol (OA) were observed in Beijing both in winter and summer
(Hu et al., 2016), based on measured result (Yu et al., 2019) from transmission electron microscopy, up to
74 % by a number of non-sea-salt sulfate particles were coated with organic matter (OM). The organic
coating can effectively reduce the reactive sites in the surface of particles hence reduce the reaction
probability of $SO_2$ with surface metal. In the other hand, the widespread presence of aerosol organic
coating can also influence the bulk $SO_2$ catalysed by aqueous TMI but not only the surface reactions. This
effect is mainly achieved by the change of $SO_2$ solubility and diffusion coefficient rather than the rates of
catalytic reactions with TMI. Although the solubility of $SO_2$ in organic solvent changes a lot with the
component of organic (Zhang et al., 2013; Huang et al., 2014a), according to previous studies of $SO_2$
uptake coefficient with sea-salt aerosol (Gebel et al., 2000) and secondary organic aerosol (SOA) (Yao et
al., 2019), no obvious uptake coefficient reduction was observed with the organic coating further proving
the minor influence of the organic coating on bulk reaction rates. The catalytic reaction of $SO_2$ with
aqTMI may less affected by aerosol organic coating compared to $SO_2$ with Mn-surface. For these reasons,
the surface reaction of $SO_2$ with Mn and other metals in actual aerosol conditions remain unclear with
high uncertainties and need further evaluation. The relevant calculation results of WD-14 and PKU-17 in
this paper represent the upper limit of Mn-surface contribution. The missing contribution in WD-14
polluted conditions may mainly come from organic photosensitizing molecules such as HULIS (Wang et
al., 2020) under stronger UV in summer or other SOA coupled mechanisms.
The results in this paper indicate that sulfate formation has different chemical behaviours in different
conditions. Aqueous TMI-catalysed oxidation was the most important pathway followed by the surface
oxidation of Mn in both winter and summer, while the hydroxyl and criegee radicals oxidations contribute
significantly in summer. Due to the differences in the physical and chemical properties between aerosol
water, fog water and cloud, nitrogen dioxide oxidation is the dominant pathway in higher pH range and
hydroperoxide and ozone oxidations dominated for the cloud. In model studies, the averaged and fixed
values should be used dialectically and carefully in the calculation of sulfate formation rate because of the
mutual restriction between factors such as pH, effective ion activity and concentration, and aerosol water
content. Model evaluation or numerical calculations of secondary pollutants should focus on the
application of actual atmospheric conditions observed in field campaigns with the application of closure
study. Our results highlight the important role of aerosol aqTMI in sulfate formation during haze periods
and the monitoring network of aerosol metal is necessary for the studies of secondary sulfate formation.
The aqTMI independent of solar radiation also explains the explosive growth of sulfate production at
night-time, which is frequently observed during haze episodes in the NCP.
Compared to the gas-phase oxidants, the control of anthropogenic emissions of aerosol TMI is conducive
to the reduction of secondary sulfates. The promotion of clean energy strategies aiming at reducing coal
burning and vehicle emissions to improve air quality in North China has reduced not only the primary
emissions of $SO_2$ but also the anthropogenic emissions of aerosol TMIs (Liu et al., 2018) and thus the
production of secondary sulfate. What' more, China's ecological and environmental protection measures
for tree planting and afforestation are conducive to reducing the generation of dust especially in the spring
can further reducing the quality of metal Fe concentrations in aerosols.
Our findings showed that urban aerosol TMIs contribute to sulfate formation during haze episodes and
play a key role in developing mitigation strategies and public health measures in megacities worldwide,
but the physicochemical processes of transition metals in particles require further research. Dissolved Mn
concentrations in this study were estimated based on previous studies. The solubility of transition metals
in aerosol water varying largely due to several factors including various source emissions, aerosol organic
matter and pH (Paris and Desboeufs, 2013; Wozniak et al., 2015; Tapparo et al., 2020) were not fully
considered in this study. Influences of organic matter and photosensitizing molecules on the solubility of
transition metal and the mechanism of sulfate formation need further research to understand this complex
and dynamic multiphase process from a broader perspective.

# Methods

## M. 1 Sampling location and experimental methods

The data from the 2014 Wangdu (WD) and 2017 Peking University (PKU) field campaigns, both
conducted in summer, were used in our analysis. The WD field campaign was carried out from June to
July 2014 at a rural site in Hebei (38.70° N, 115.15° E) characterized by severe photochemical smog
pollution (Tan et al., 2017; Song et al., 2020). The 2017 PKU campaign was performed from November
to December 2017 at the campus of Peking University (39.99° N, 116.31° E), which is in the city centre
of Beijing and characterized by strong local anthropogenic emissions from two major roads (Ma et al.,

372     2019).

Observations from both field campaigns include gas-phase measurements of $SO_2$ and $O_3$ from commercial
Thermo Scientific monitors and $NO_2$ detected after conversion through a custom-built photolytic
converter with UV-LED at 395 nm; aerosol number concentration and distribution from a set of
commercial particle instruments containing Nano scanning mobility particle sizer (SMPS) and
aerodynamic particle sizer (APS) to cover the size range of 3 nm to 10 μm. The mass concentration of
$PM_{2.5}$ was measured by commercial Ambient Particulate Monitor (TEOM).  The In-situ Gas and Aerosol
Compositions monitor (IGAC) (Young et al., 2016), which can collect gases and p articles
simultaneously, was used to measure water-soluble ions online with 1-h time resolution. Both gas and
aerosol samples were injected into 10 mL glass syringes, which were connected to an ion chromatograph
(IC) for analysis (30-min time resolution for each sample). The concentrations of eight water soluble
inorganic ions ($NH_4^+$, $Na^+$, $K^+$, $Ca^{2+}$, $Mg^{2+}$, $SO_4^{2-}$, $NO_3^-$, and $Cl^-$) in fine particles were measured.
Transition metal (Fe and Cu) concentrations in $PM_{2.5}$ were measured using the Xact 625 Ambient Metal
Monitor. With Xact, ambient air was introduced through a $PM_{2.5}$ cyclone inlet at a constant flow rate of
16.7 L min$^{-1}$ and collected on the reel-to-reel poly tetrafluoroethylene filter. Then trace elements in
ambient fine particles on the filter were automatically detected using the United States Environmental
Protection Agency (USEPA) standard method via x-ray fluorescence (XRF) analysis (Gao et al., 2016;
Zhang et al., 2019). Ambient temperature and pressure data were measured using commercial
meteorological sensors; selected volatile organic compounds (VOCs) were measured via off-line gas
chromatography–mass spectrometry (GC-MS) in tower measurements using sampling canisters and via
online GC−MS in the surface campaign. The OH and $HO_2$ concentrations were measured via laser-
induced fluorescence (LIF) with the time resolution of 30 s as described in previous study (Ma et al.,
2019). The concentrations of gas-phase peroxides were measured using high-performance liquid
chromatography (HPLC, Agilent 1200, USA) with a time resolution of 21 min.

## M. 2 Brief overview of the PKU-MARK model

The Multiple-phAse Reaction Kinetic Model (PKU-MARK) was first developed to calculate the
heterogeneous reaction rate of reactive gas molecules (Song et al., 2020). The units of aqueous reagents
are converted to molecules·cm$^{-3}$ in the model by a factor $k_{mt}$, which combines both gas-phase molecular
diffusion and liquid-phase interface mass transport processes (Schwartz, 1984; Schwartz, 1986) and used
in the calculation for gas–liquid multiphase reactions in many modelling studies (Lelieveld and Crutzen,
1991; Chameides and Stelson, 1992a; Sander, 1999; Hanson et al., 1994; Song et al., 2020). In this study,
the PKU-MARK model was further developed with the correction of ionic strength for all ions and
reactants and applied to a size-segregated system to investigate the influence of aerosol particle size
distribution and ALWC distribution. Eleven bins of aerosol particle diameters and corresponding ALWC
values were applied in the model. With the input of one-hour averaged parameters observed in the field
campaign, the PKU-MARK model produced the state-state concentrations of aqueous reactants including
reactive oxygen species ($H_2O_2$, $O_3$, OH, $HO_2$, $O_2^-$), Fe (III), Mn (II), $SO_{2(aq)}$, and $NO_{2(aq)}$. Considering the
mutual influence of various factors in the reaction system can effectively prevent bias caused by
arbitrarily fixing a certain value as was often done in previous studies.

## M. 3 Calculation of aerosol pH, aerosol liquid water, and ionic strength

ALWC and aerosol pH were calculated using the ISORROPIA-II model and measured concentrations of
inorganic ions in particles. ISORROPIA-II is a thermodynamic equilibrium model that predicts the
physical state and composition of atmospheric inorganic aerosols. Its ability to predict pH has been
demonstrated in detail in previous studies (Guo et al., 2015; Xu et al., 2015). Ionic strength was calculated
via equation (2) (Ross and Noone, 1991):
$$I_s = \frac{1}{2} \cdot \sum m_i \cdot z_i^2, \tag{2}$$

where $m_i$ is the molality of an ion (mol L$^{-1}$), and $z_i$ is the corresponding charge. In the PKU-MARK
model, reaction rates were replaced by the activity coefficient. The ionic strength was estimated using the
ISORROPIA-II model assuming that the condensed phase is in the meta-stable state and complete
external mixing state.
In order to consider the influence of particle diameter on aqueous $SO_2$ concentrations, which is key to
calculate sulfate formation, we used a 11-bin actual particle diameter distribution rather than one even
distribution used in previous studies (Cheng et al., 2016). The distribution of particle number
concentration and water content is illustrated in **Fig. S2**. We also considered the distribution of ALWC in
different particle diameter bins based on the κ–Köhler theory (Petters and Kreidenweis, 2007) using
observed kappa values from High Humidity Tandem Differential Mobility Analyser (HH-TDMA) and the
Twin Differential Mobility Particle Sizer (TDMPS)/APS (Bian et al., 2014). Calculated ALWC values
were strongly correlated with the ISORROPIA-II results (**Fig. S3**).
To combine both gas-phase molecular diffusion and liquid-phase interface mass transport processes, the
approach adopted in this study uses one variable called $k_{mt}$ (Schwartz, 1984; Schwartz, 1986), which is
used in multiphase reactions in many modelling studies (Lelieveld and Crutzen, 1991; Chameides and
Stelson, 1992b; Sander, 1999; Hanson et al., 1994). The definition of $k_{mt}$ is given in equation (3):
$$k_{mt} = \left(\frac{R_d{}^2}{3D_g} + \frac{4R_d}{3\upsilon_{HO_2}\alpha}\right)^{-1}. \tag{3}$$

The rate of gas-phase reactions (*X*) diffusing and dissolving to the condensed phase can be calculated in
the framework of aqueous-phase reactions as $k_{mt\_X} \times ALWC$ where *X* is the reactant molecule (please see
**Table S8** for more details). Moreover, the conversion rate of aqueous-phase reactions to gas-phase
reactions can be calculated as $\frac{k_{mt\_X}}{H^{cc} \times RT}$. The unit of $k_{mt}$ is s$^{-1}$, as $k_{mt}$ contains the conversion from m$_{air}^{-3}$ of
the gas-phase molecule concentrations to m$_{aq}^{-3}$ of the aqueous-phase molecule concentrations. Particle
diameter can influence the mass transport rate of $SO_2$ and its aqueous concentration. Based on the model
results of (Xue et al., 2016), diameter had an impact on sulfate formation rates: for larger particles
(radius >1 μm), $k_{mt}$ is determined by gas-phase diffusion; for smaller particles (radius <1 μm), $k_{mt}$ is
determined by the accommodation process. The PKU-MARK model can simultaneously simulate two-
phase (gas and liquid) reaction systems in the same framework.

## M. 4 Model Evaluation

Concentrations of sulfate were calculated by integrating an extension of the Eulerian box model described
in previous study (Seinfeld and Pandis, 2016). Sulfate concentrations are related to dry deposition,
transport, dilution as the boundary layer height (BLH) expands, emissions, and net production. Due to the
higher and more dramatically diurnal changing BLH in summer (Lou et al., 2019), and the lack of
relevant data in WD-14 field campaign, we could not get the modelled results of sulfate concentrations in
summer haze periods. Direct emissions and transport of sulfate were not considered in the calculation
because secondary sulfate is the predominant source in winter haze periods. Dilution was not considered
either because the atmosphere is relatively homogeneous during winter haze episodes. Since haze events
are normally accompanied by a low boundary layer height ($H_t$), $H_t$ was set at 300 m at night-time and 450
m at noon (Xue et al., 2016). At other times, $H_t$ was estimated using a polynomial (n = 2) regression as
recommended in previous study (Xue et al., 2016). The diurnal trends of sulfate concentrations of the
winter haze period using the deposition velocity of 1.5 cm/s and of 2 cm/s in summer are shown in **Fig. 1**
**(c) and (d)**. Model results had the same trend with the observed values and could explain the missing
source of sulfate aerosol to some extent in winter while with high uncertainties in summer condition.


 **Table 1. Averaged results of observed meteorological parameters, trace gases concentrations**

 **transition metal concentrations such as Fe, Cu, Mn and calculated ALWC, ionic strength, pH and**

 **sulfate formation rates in different pollution conditions in two field campaigns ($\pm 1\sigma$).**

| Parameters | Clean | Slightly polluted | Polluted | Highly polluted |
|---|---|---|---|---|
| **Winter** | | | | |
| RH (%) | $25.0\pm8.3$ | $37.1\pm11.5$ | $44.8\pm11.9$ | $63.6\pm19.5$ |
| Temperature (K) | $273.0\pm4.6$ | $274.1\pm3.3$ | $273.6\pm2.6$ | $273.8\pm2.3$ |
| $SO_2$ (ppbV) | $2.4\pm1.4$ | $5.8\pm2.0$ | $6.5\pm2.6$ | $5.5\pm3.0$ |
| $NO_2$ (ppbV) | $21.1\pm10.4$ | $37.6\pm6.3$ | $44.1\pm6.1$ | $57.6\pm8.7$ |
| OH (#/cm$^3$) | $(4.67\pm3.73)\times10^5$ | $(5.02\pm5.22)\times10^5$ | $(4.42\pm2.78)\times10^5$ | $(4.36\pm3.06)\times10^5$ |
| $H_2O_2$ (pptV) | $29.8\pm20.8$ | $23.5\pm27.2$ | $19.5\pm39.6$ | $20.9\pm22.8$ |
| $O_3$ (ppbV) | $14.8\pm11.9$ | $3.2\pm5.7$ | $2.1\pm2.7$ | $1.1\pm1.2$ |
| $SO_4^{2-}$ ($\mu$g/m$^3$) | $3.5\pm1.5$ | $6.4\pm3.5$ | $8.3\pm4.2$ | $16.6\pm6.6$ |
| Fe (ng/m$^3$) | $348.4\pm263.0$ | $564.2\pm188.2$ | $725.5\pm258.6$ | $1300.6\pm289.5$ |
| Cu (ng/m$^3$) | $7.0\pm5.0$ | $13.8\pm4.2$ | $18.7\pm6.0$ | $29.3\pm6.6$ |
| Mn (ng/m$^3$) | $12.4\pm9.4$ | $20.1\pm6.7$ | $25.9\pm9.2$ | $46.5\pm10.3$ |
| ALWC ($\mu$g/m$^3$) | $3.1\pm2.6$ | $3.8\pm4.4$ | $11.9\pm15.6$ | $82.4\pm67.3$ |
| Surface area ($\mu$m$^2$/cm$^3$) | $263.2\pm171.5$ | $714.3\pm242.2$ | $1253.3\pm448.9$ | $2628.6\pm1164.4$ |
| $PM_{2.5}$ ($\mu$g/m$^3$) | $18.3\pm10.1$ | $52.0\pm10.0$ | $101.7\pm18.2$ | $190.0\pm30.0$ |
| pH | $4.43\pm1.12$ | $4.52\pm0.76$ | $4.93\pm0.57$ | $4.77\pm0.39$ |
| Ionic Strength (M) | $170.34\pm88.32$ | $89.32\pm55.19$ | $61.59\pm38.7$ | $36.27\pm36.93$ |
| d[S(VI)]/dt ($\mu$g/m$^3$/h) | $1.3\pm1.88$ | $2.25\pm2.15$ | $2.35\pm2.19$ | $3.98\pm2.75$ |
| **Summer** | | | | |
| RH (%) | $69.5\pm17.9$ | $64.4\pm18.4$ | $66.4\pm13.0$ | $65.6\pm7.7$ |
| Temperature (K) | $296.5\pm3.6$ | $298.5\pm4.4$ | $299.1\pm2.9$ | $298.9\pm3.1$ |

| | | | | |
|---|---|---|---|---|
| $SO_2$ (ppbV) | 2.4±2.0 | 4.6±4.4 | 5.6±5.0 | 7.9±4.0 |
| $NO_2$ (ppbV) | 8.7±4.9 | 9.6±5.6 | 9.0±5.5 | 12.3±6.1 |
| OH (#/cm$^3$) | $(2.38\pm2.44)\times10^5$ | $(3.27\pm3.21)\times10^5$ | $(2.77\pm2.26)\times10^5$ | $(3.50\pm3.38)\times10^5$ |
| $H_2O_2$ (pptV) | 466.2±571.6 | 355.5±488.0 | 596.1±777.0 | 173.6±348.6 |
| $O_3$ (ppbV) | 46.0±30.3 | 50.9±30.6 | 53.0±26.6 | 48.5±28.5 |
| $SO_2^{4-}$ (μg/m$^3$) | 7.2±2.6 | 11.0±4.9 | 17.8±6.0 | 24.4±6.0 |
| Fe (ng/m$^3$) | 521.6±286.6 | 469.3±151.7 | 535.2±177.0 | 730.9±156.6 |
| Cu (ng/m$^3$) | 26.6±18.8 | 37.7±31.8 | 33.8±26.0 | 47.1±36.3 |
| Mn (ng/m$^3$) | 18.6±10.2 | 16.8±5.4 | 19.1±6.3 | 26.1±5.6 |
| ALWC (μg/m$^3$) | 31.8±30.9 | 35.7±32.8 | 48.6±31.4 | 58.8±14.4 |
| Surface area (μm$^2$/cm$^3$) | 767.8±265.6 | 925.0±213.9 | 1389.0±312.6 | 1711.1±729.6 |
| $PM_{2.5}$ (μg/m$^3$) | 20.1±10.2 | 54.9±11.7 | 104.8±20.5 | 194.6±32.9 |
| pH | 4.48±0.48 | 4.19±0.66 | 4.17±0.48 | 4.33±0.44 |
| Ionic Strength (M) | 20.04±17.53 | 25.44±20.83 | 24.27±14.06 | 24.2±9.19 |
| d[S(VI)]/dt (μg/m$^3$/h) | 2.13±2.03 | 3.81±4.22 | 3.79±5.66 | 5.6±4.45 |

The concentration of Mn was estimated based on the ratio of Fe/Mn observed in urban Beijing in the
literatures (summarized in **Table S9**). All mentioned aerosol data is particle matters diameter smaller than
2.5 μm, and $PM_{2.5}$ refers to the dry mass concentration of fine particulate matters.

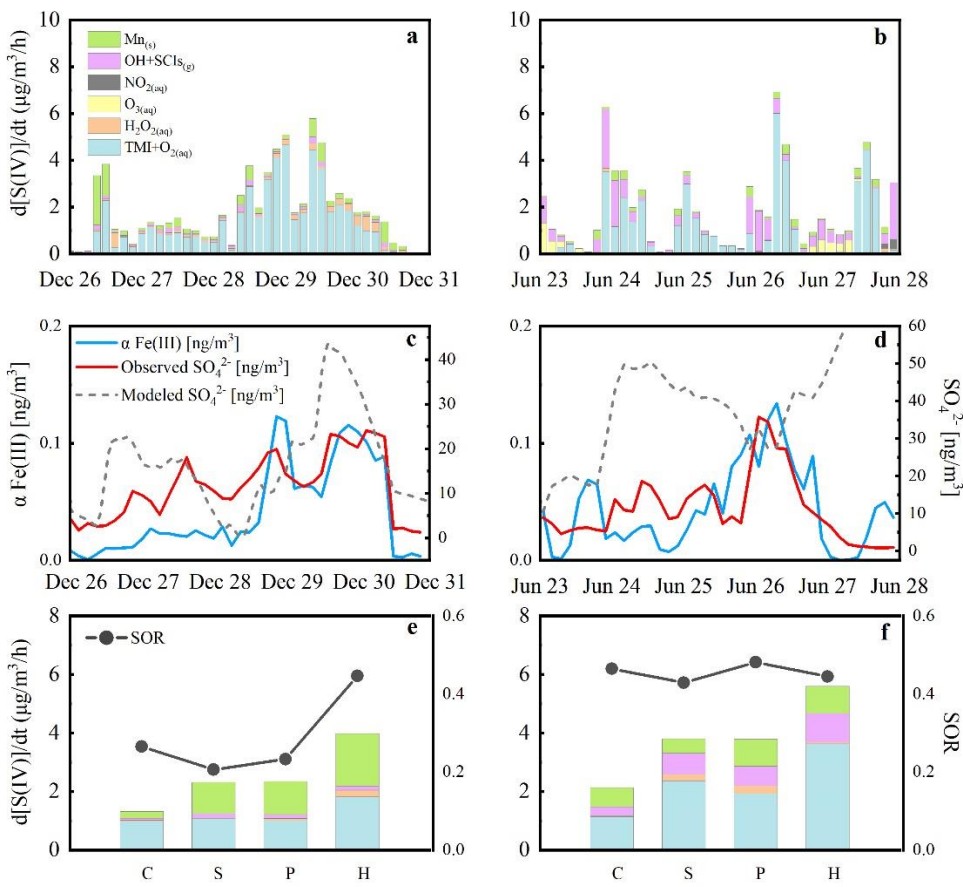


**Figure 1. Three-hour average sulfate formation rates during haze periods in winter and summer (a)&(b), corresponding effective Fe (III) concentrations and sulfate concentrations (c)&(d), sulfate formation rates (the histogram) and SOR (the dotted lines) in different pollution levels in two field campaigns (e)&(f).**

The contributions to sulfate formation from each multiphase oxidant pathways including Mn-surface oxidant (green), gas phase OH radical and Stabilized Criegee Intermediates (SCIs) oxidant (pink), aqueous phase $NO_2$ (grey), $O_3$ (yellow), $H_2O_2$ (orange) and aqTMI (blue) were coloured in the figure. Obvious particle growth and removal was observed in winter (26[th] to 31[st], December, 2017) and diurnal variation patterns of sulfate concentration were observed in summer (23[th] to 28[th], June, 2014). Diurnal trends of modelled winter period's sulfate concentration (grey dash line) using deposition velocity as 1.5 cm/s in winter and 2 cm/s in summer are illustrated in panel (c) and (d). The dotted lines in the (e), winter

and (f), summer indicate the SOR with pollution level in the whole campaigns and the capitalized letters
"C", "S", "P", "H" are the abbreviations for "Clean", "Slightly polluted", "Polluted" and "highly
polluted", respectively.

**Figure 2. Multiphase sulfate production under actual ambient conditions (a, c) and averaged**
**conditions (b, d) in winter (a, b) and summer (c, d) in the North China Plain.**

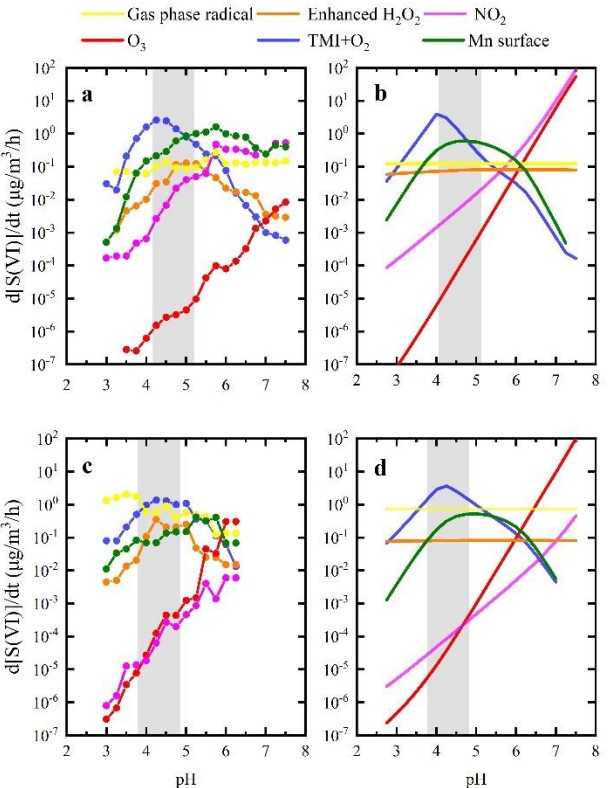


Given the actual measured concentration, the steady-state concentration of each reactant was calculated
using the MARK model accounting for the impact of ionic strength on the Henry's law coefficient of the
gas-phase reactants. Panels (a) and (c) show the cluster averaged results with a pH span of 0.5. Panels (b)
and (d) show the sulfate formation rate obtained by fixing the average precursors levels during the haze
periods and by changing the aerosol pH, which is consistent with the calculation method of  previous
studies (Cheng et al., 2016). Grey-shaded areas indicate the ISORROPIA-II (Fountoukis and Nenes,
2007) model calculated pH ranges during the haze periods of two field campaigns. The coloured solid
lines represent sulfate production rates calculated for different multiphase reaction pathways with
oxidants: enhanced $H_2O_2$, $O_3$, TMIs, $NO_2$, surface Mn and gas-phase radicals (OH+SCIs). The solid
orange line represents the calculated sulfate formation rate via H2O2 with a factor of 100 in winter and
summer according to the latest research results (Liu et al., 2020b). Reactant concentrations, aqueous
reaction rate expressions, and rate coefficients are summarized in the SI.
**Figure 3. Variation of PM$_{2.5}$, ionic strength, aerosol pH, particle specific surface areas and sulfate**
**formation rates from different pathways with aerosol liquid water content (ALWC) during winter**
**field campaign**.

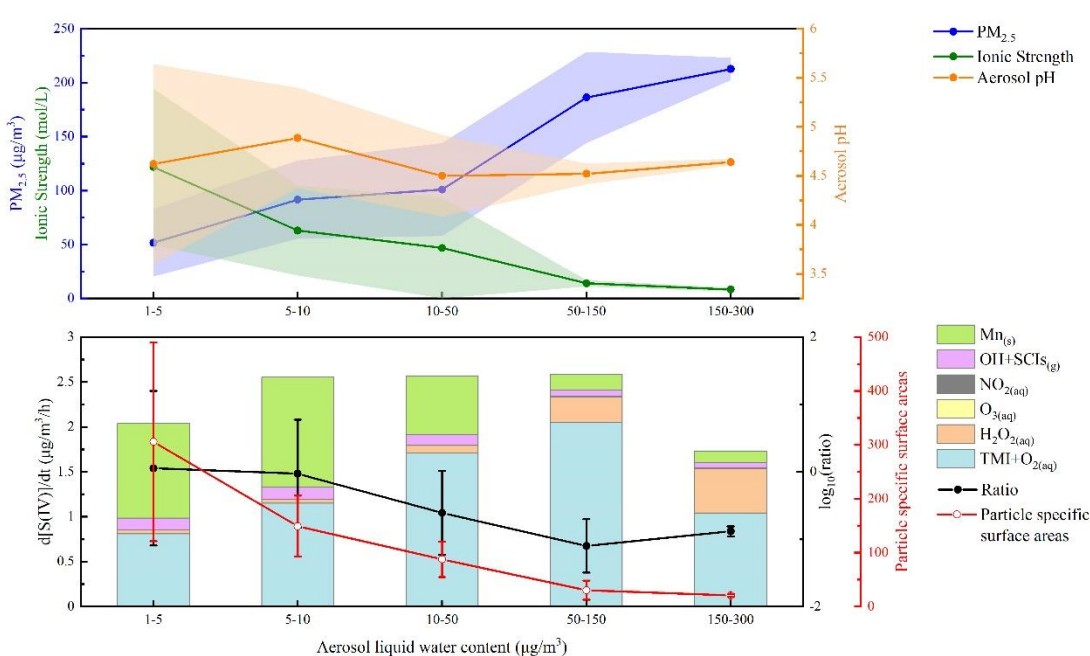

The total number of valid data points shown in the figure is 479. The shaded area refers to the error bar
($\pm 1$ $\sigma$) of PM$_{2.5}$ mass concentration, aerosol ionic strength and pH calculated by ISORROPIA-
II(Fountoukis and Nenes, 2007). Ratio in the second panel refers to the ratio of contributions from Mn-
surface to aqTMI to produce sulfate. Particle specific surface areas represent the ratio of particle surface
area ($\mu m^2/cm^3$) and mass concentration ($\mu g/m^3$).


**Figure 4. Bar graph showing modelled contributions of various pathways to sulfate formation**
**under different pollution conditions.**

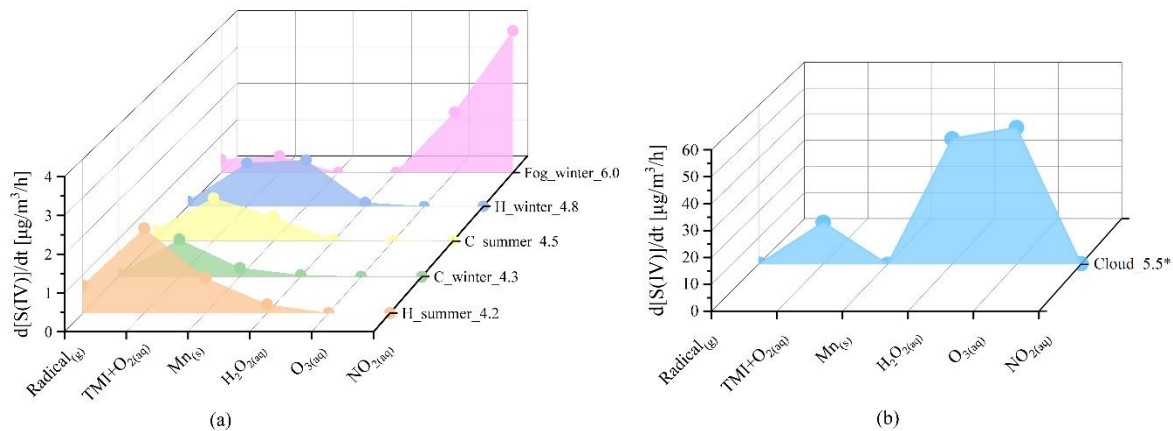


Different pollution conditions including clear ($PM_{2.5}$ smaller than 35 μg/m³) in winter PKU 2017
(C_winter_4.3) and summer WD 2014 (C_summer_4.5); pollution ($PM_{2.5}$ larger than 75 μg/m³) in PKU
2017 (H_winter_4.8), WD 2014 (H_summer_4.2); fog conditions used in a previous study (Xue et al.,
2016) (Fog_winter_6.0) and cloud conditions (Cloud_5.5) simulated by Seinfeld and Pandis (2016). The
number in each label indicates the average pH value chosen in these calculations. We assumed that the
cloud water content is 0.1 g/m³ in the last condition, and reduced the $H_2O_2$ concentration to 0.1 ppb
compared to the high value used before (Seinfeld and Pandis, 2016).

## Data Availability

Data supporting this publication are available upon request for the corresponding author

(k.lu@pku.edu.cn).

## Conflict of interests

The authors declare that they have no conflict of interest.

## Acknowledgements

This study was supported by the by the National Key Research and Development Program of China

(2019YFC0214800), the Beijing Municipal Natural Science Foundation for Distinguished Young

Scholars (JQ19031), the National Key Research Program for Air Pollution Control (DQGG202002), the

National Natural Science Foundation of China (21976006).

## Author Contributions

Keding Lu conceived the study. Huan Song and Keding Lu developed the MARK model for multiphase

simulations. Can Ye provide supports in the calculation. Huan Song performed the model simulations and

wrote the manuscript with Keding Lu and Can Ye. Keding Lu and Yuanhang Zhang lead the two field

campaigns. Keding Lu, Huabin Dong, Shule Li, Shiyi Chen, Zhijun Wu, Mei Zheng, Limin Zeng, Min Hu

& Yuanhang Zhang provide campaign data for the analysis.

## Supplementary Information

Supplementary information includes:

- Supplementary Information Text

  Text S1. Activity coefficients of main reactants in the MARK model

  Text S2. The concentration of aerosol particle transition metals in urban areas

- Supplementary Information Figures Fig.S1-S9

  Fig. S1. Ionic strength of aerosol particle solution influence on the aqTMI rate constant.

Fig. S2. Distribution of ALWC and number concentration with aerosol particle bins in two
campaigns.
Fig. S3. Calculated aerosol water by ISORROPIA-II model and H-TDMA method in two field
campaigns during haze periods. The plots were coloured with the relative humidity values. The
black dashed line in the figure is the 1:1 baseline, and the red solid line is the linear fitting result
assuming the intercept is zero.
Fig. S4. Time series of observed gas-phase pollutants concentrations, RH, Temperature, PM2.5
mass loading and calculated aerosol pH and water content and sulfate formation rates in these
four haze periods in PKU-17 field campaign.
Fig. S5. SOR ($\equiv n(SO_2)/n(SO_2+SO_4^{2-})$) correlations with effective Fe (III) concentrations in
PKU-17 winter field campaign.
Fig. S6. SOR ($\equiv n(SO_2)/n(SO_2+SO_4^{2-})$) correlations with odd oxygen ($[O_x]\equiv[O_3]+[NO_2]$) and
relative humidity (RH) in WD-14 summer field campaign
Fig. S7 the "dilution effect" of Fe mass concentration and ALWC increasing with PM mass in
winter and summer.
Fig. S8. Variation of PM2.5, ionic strength, aerosol pH, particle specific surface areas and
sulfate formation rates from different pathways with aerosol liquid water content (ALWC)
during summer field campaign.
• Supplementary Information Tables S1-S9.
Table S1. Reaction rate expression and constant for SO2 oxidation by OH in the gas-phase.
Table S2. Aqueous-phase reaction rate expressions, rate constants (k) and influence of ionic
strength (Is) on k for sulfate production in aerosol particle condensed phase.
Table S3. Calculations of Henry' law coefficients and influence of ionic strength.
Table S4. Typical activity coefficient values and expressions used in the MARK model.
Table S5. Kinetic data for the simulation of reactions in the aerosol particle condensed phase.

569          Table S6. Photolysis rates (aqueous phase) used in the model at noon (sza = 20°)

570          Table S7. Aqueous equilibrium reactions

571          Table S8. Kinetic data for the simulation of gas-liquid phase conversion reactions

572          Table S9. Concentration of transition metals in PM2.5 in urban areas.

573     •   SI References

574

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
