# Peer review of "A Comprehensive Observational Based Multiphase Chemical Model Analysis of the Sulfur Dioxide Oxidations in both Summer and Winter"

_Atmospheric Chemistry and Physics, 2021_

## Referee Comment (RC2)

The manuscript entitled "A Comprehensive Observational Based Multiphase Chemical Model Analysis of the Sulfur Dioxide Oxidations in both Summer and Winter" by Song et al. presents the comprehensive evaluation of the contribution of different sulfate production pathways in both summer and winter by their self-developed multiphase box model (PKU-MARK). The model includes nearly all the established sulfate production pathways, providing valuable insights into the sulfate formation evaluation. Overall, I have some concerns that need the authors to clarify before that I can recommend publication in *Atmospheric Chemistry and Physics*.

1. The concentration of TMIs is vital in this work since the two dominant sulfate production pathways the authors proposed are aqTMI and Mn-surface. An online monitor measured the Fe and Cu concentrations. Due to the lack of Mn data, the authors propose a fixed ratio of Fe/Mn to stimulate the concentration of Mn. So how about the uncertainty of this method? It is better to compare the concentration of Mn with literature results in the same region.

2. The authors state that the average soluble percentage of Fe and Mn in winter polluted conditions was 0.79% and 19.83%. However, the water-soluble fraction of Fe and Mn may change a lot in different regions, as stated in the manuscript. Also, the solubility may change under clean conditions and polluted conditions. It is better to add some discussion about the sensitivity of the solubility of Fe and Mn to the model results.

3. The authors declare that their result is consistent with the result of the WRF-CHEM study. However, in the cited work, the ionic strength inhibition effect was not included. More discussion about the results is needed.

4. It is important that the model has considered the activity coefficient values and reactions about oxalate and Fe. So how about the concentration of oxalate used in the model?

Other minor comments:

Line 11, the wording of sulfate should be better consistent, "sulphate" or "sulfate".

Line 13, the statement of "observed concentrations of transition metal ions" is not appropriate from my perspective, given the authors only measured the total concentration of Fe and Cu.

Line 22, "…affect the environmental quality and human health", references to support this conclusion are lacking.

Line 149, "Obvious correlations between alpha-Fe(III) and sulfate…", The author may better calculate the $R^2$ of alpha-Fe(III) and sulfate.

Line 380, in Fig. 1(d), the modeled sulfate concentration line is missing.

---

## Author Comment (AC1)

**Response to the comments of referee #1**

Comment:
This paper focuses on evaluating various pathways for conversion of SO2 to PM2.5 sulfate over the North China Plane, a topic that has generated a large number of published papers in the last 2 years or so. The modeling work here is likely the most comprehensive analysis comparing all current mechanisms. Whether the results are accurate or not is hard to assess, but this work does provide valuable new insights by contrasting all the possible (main) mechanisms. The paper is appropriate for publication in this journal; however, it could be improved by careful editing focusing on grammar and clearer explanations. Additional broad and specific comments are provided below.

We thank the reviewer for the helpful comments. The referee's comments are first given in black type, followed by our response to each in turn in blue type. Any changes to the manuscript in response to the comments are then given in quotation marks in red type.

Broad Comments:

1 It would be worthwhile stating somewhere (maybe include in Table 1) what fraction of the $PM_{2.5}$ mass is sulfate; ie, how important is this problem.
We added the mass concentration on different pollution level in Table 1. During haze periods in winter and summer field campaigns mentioned in the MS, the contribution of $SO_4^{2-}$ mass to $PM_{2.5}$ dry mass is in the range of 4%-59%, with an average value as 15% in winter and in the range of 1%-52%, with an average value as 19% in summer. In the introduction part, we have cited the literatures to illustrate the importance of secondary sulfate during haze periods, thus we added the fractions data in Table 1 in the revised MS.

**Table 1. Averaged results of observed meteorological parameters, trace gases concentrations transition metal concentrations such as Fe, Cu, Mn and calculated ALWC, ionic strength, pH and sulfate formation rates in different pollution conditions in two field campaigns (±1σ).**

| Parameters | Clean | Slightly polluted | Polluted | Highly polluted |
|---|---|---|---|---|
| Winter | | | | |
| … | | | | |
| $PM_{2.5}$ (µg/m$^3$) | 18.3±10.1 | 52.0±10.0 | 101.7±18.2 | 190.0±30.0 |
| … | | | | |

| | Summer | | | |
|---|---|---|---|---|
| … | | | | |
| PM$_{2.5}$ (µg/m$^3$) | 20.1±10.2 | 54.9±11.7 | 104.8±20.5 | 194.6±32.9 |
| … | | | | |

The concentrations of Mn were estimated based on the ratio of Fe/Mn observed in urban Beijing in the literatures (summarized in **Table S9**). All mentioned aerosol data is particle matters diameter smaller than 2.5 µm and PM$_{2.5}$ refers to the dry mass concentration of fine particulate matters.

2 These authors find that the most important route involves transition metal ions, however the concentration of these species seems to be very uncertain since in this work only the total (elemental) concentration was measured and the actual TMI species concentrations had to be estimated based on estimated solubilities (which can vary over a large range). This substantial uncertainty needs to be addressed. For example, maybe the authors should provide a range in predicted sulfate formation rates for the TMI route, include this in the plots (say something similar to Fig 2, if possible) and discuss the implications (does it change the findings).

We added the following discussion of transition metal sensitivity on sulfate formation in PKU-17 winter field campaign in the revised SI Text S4, Figure S9 and Table S10.

Water soluble fraction of Fe, Mn and Cu can vary over a large range. A large part of the soluble metals is in the form of soluble organic complexes or hydroxides rather than ions in aerosol particles. There are evidences that the existence of various aerosol water soluble organic acids (oxalate, malonate, tartrate and humid acid) cause an enhancement of Fe, Cu and Mn solubility and the formation of metal-organic complex (Paris and Desboeufs, 2013; Wozniak et al., 2015; Tapparo et al., 2020). What's more, the dissolution of Fe and Mn is highly influenced by aerosol pH. Circumneutral pH leads to a supersaturated soluble Fe (III), which then precipitates out of the solution. For these reasons, the promotion of metal solubility may have non-proportional influences on the aqueous concentration TMI. We conducted the sensitivity analysis for the solubility of Fe from 1% to 15% (Scenario one with fixed aqueous Mn and Cu concentration consist with the base run in the MS, Scenario two with fixed ratio of soluble Fe/Mn and Fe/Cu mass, ie, Mn solubility in the range of 10% to 100%, Cu in 5% to 75%, as shown in Table S10). Other aerosol component concentration, ionic strength, ALWC, observed meteorological parameters and trace gases concentrations stay consistent with the base run.

[Figure]

Figure S.9 Sensitivity analysis of transition metal including Fe, Mn and Cu solubility influences on the averaged sulfate formation rates in PKU-17 field observation. Dotted lines in the figure show the cluster averaged results with a pH span of 0.5 under actual ambient conditions with different transition metal solubilities.

Table S10. Base run and scenarios of the solubility sensitivity analysis.

| Sensitivity Analysis | Solubility of transition metals | Sulfate formation contribution in haze pH range (4.2-5.2) ($\mu g/m^3/h$) |
|---|---|---|
| Base Run | 5% Fe + 50% Mn + 25% Cu | 0.80 - 2.58 |
| Scenario 1 | 15% Fe + 50% Mn + 25% Cu | 3.49 - 8.57 |
| | 1% Fe + 50% Mn + 25% Cu | 0.05 - 0.16 |
| Scenario 2 | 15% Fe + 100% Mn + 75% Cu | 12.97 - 32.87 |
| | 1% Fe + 10% Mn + 5% Cu | 0.009 - 0.004 |

As shown in Figure S.9, In the range of winter haze periods pH (4.2-5.2), averaged sulfate formation rates in PKU-17 field observation is non-proportional to the initial transition metal solubility. Fe solubility increasing from 1% to 5% will cause d[S(VI)]/dt to increase over an order of magnitude, and increasing to 15% cause no obvious effect when pH smaller than 4.2, while obvious effect the pH ranging from 4.2 to 6. This phenomenon may be due to the piecewise calculation equations of TMI-catalysis oxidizing $SO_2$ as mentioned in the SI and following. In the presence of TMI

organic complexes and redox reactions, this equation may need to be further verified, but verification is not within the scope of this study. It is obvious that the d[S(VI)]/dt changes caused by the proportional expansion of the solubility of the three transition metals (Scenario 2) is more significant especially when the solubility is reduced to 1%+10%+5%. Increasing of solubility to 15%Fe+100%Mn+75%Cu can increase sulfate formation rate to 5-84 times higher than in base run during haze periods pH as 4.2-5.2. This can explain to a certain extent that excessive TMI concentrations will not cause a sharp increase in d[S(VI)]/dt, which may be due to the buffering effect caused by the formation of organic complexes.

Part of Table S2. Aqueous-phase reaction rate expressions, rate constants (k) and influence of ionic strength (Is) on k for sulfate production in aerosol particle condensed phase.

| Oxidants | The reaction rate expressions ($R_{S(IV)+oxi}$), constants ($k$) and influence of $I_s$ (in unit of M) on $k$ [a] | Notes | References |
|---|---|---|---|
| TMI+O$_2$ [f] | $k_6[H^+]^{-0.74}[S(IV)][Mn(II)][Fe(III)]$ (pH $\leq$ 4.2) $k_6 = 3.72\times10^7\times e^{(-8431.6\times(1/T-1/297))}$ M$^{-2}$ s$^{-1}$ $k_7[H^+]^{0.67}[S(IV)][Mn(II)][Fe(III)]$ (pH > 4.2) $k_7 = 2.51\times10^{13}\times e^{(-8431.6\times(1/T-1/297))}$ M$^{-2}$ s$^{-1}$ | | Ibusuki and Takeuchi (1987) |
| | $\log_{10}(\frac{k}{k^{I_s=0}}) = \frac{b_4\sqrt{I_s}}{1+\sqrt{I_s}}$ [g] $b_4$ is in range of –4 to –2 | $I_{s, max} =$ 2 M | Martin and Hill (1987, 1967) |

3 The concentration of the TMIs (mainly Fe(II)+Fe(III)) and Mn involved in the surface reaction chemistry determines how fast sulfate is formed (line 97 notes that the TMI concentration is crucial…). But what are the TMI concentrations, only total metals measured by XRF are given? What is unique about these regions that makes these metal ions a major route? The authors point to the haze reducing photochemistry (during pollution events the PM2.5 mass is very high compared to many other regions globally), high RH, moderate particle pH, but what about the concentration of TMI? My rough analysis suggests that the mass ratio of TMI to sulfate is much higher in this region than many others, which would also be an important reason why this route may be important in this specific region. It would also support the conclusions of the paper, that emissions of these metals should be reduced (although as I note below, I believe more details are needed on the sources of the TMI, I don't think it is solely coal combustion based on the cited paper). I think the authors should assess this question; are TMI a uniquely large fraction of the PM2.5 or (TMI/sulfate ratio) in this region? At the very least, please provide some form of assessment of TMI mass concentrations, (this could include for example the sum of the various forms since the speciation is highly variable, eg,

Fe(II)+Fe(II), etc), relative to PM2.5 or sulfate, ie, maybe in Table 1.

In our calculation, aerosol pH, aerosol water content and high transition metal concentrations synthetically cause the aqTMI catalysis oxidation pathway an important contributor to secondary sulfate formation in PKU-17 winter field campaign in Beijing. The mass ratio of Fe total mass to $SO_4^{2-}$ was in a high range compared to other observations in other regions, as shown in Table R1. However, the high concentration of transition metal does not mean that aqTMI play a role in sulfate formation. Proper aerosol liquid water content and aerosol pH ranging from 4 to 5.5 were the other two important factors improving the contribution percentage of aqTMI pathway. Compared to the total mass concentration of transition metal, effective aqueous TMI concentration is more relative to the sulfate formation. As shown in Figure 1 (c) and (d) in the MS, obvious correlations between αFe (III) (defined as the product of the Fe (III) activity coefficient, concentration, molecular weight (56) and aerosol liquid water content) and sulfate concentration were observed in the haze periods both in summer ($R^2$=0.63) and winter ($R^2$=0.71) and the correlation is consistent with verified the important contributions from aqTMI pathway to the sulfate formation. However, the calculation of αFe still has a large uncertainty (as discussed in the above response), so we can only compare the total Fe mass concentration and the sulfate concentration in various regions here in order to illustrate the high level of transition metal in Beijing.

The sources of transition metals including Fe, Mn and Cu is discussed in following response to comment 9.

Table R1. Fe/$SO_4^{2-}$ ratio in different regions.

| Location | Time | $SO_4^{2-}$ | Fe | Fe/$SO_4^{2-}$ | reference |
|---|---|---|---|---|---|
| PKU-17_highly polluted | 2017 | 16.6 ± 6.6 | 1.30 ± 0.30 | 78.35 | This study |
| PKU-17_polluted | | 8.3 ± 4.2 | 0.73 ± 0.26 | 87.35 | |
| Hongkong | 2001 | 12.76 ± 5.45 | 0.25 ± 0.12 | 19.59 | (Ho et al., 2003) |
| | | 15.29 ± 3.71 | 0.48 ± 0.50 | 31.39 | |
| | | 13.07 ± 5.17 | 0.19 ± 0.19 | 14.54 | |
| Lanzhou | 2014 | 12.0 ± 4.6 | 1.93 ± 0.95 | 161.08 | (Wang et al., 2016) |
| | | 7.6 ± 3.3 | 2.49 ± 1.55 | 327.24 | |
| Fujian | 2007 | 20.38 ± 5.85 | 0.58 ± 0.32 | 28.56 | (Yin et al., 2012) |
| Guangzhou | 2013 | 12.6 ± 7.6 | 0.16 ± 0.11 | 12.94 | (Lai et al., 2016) |
| Suzhou | 2014 | 17.3 ± 8.61 | 2.12 ± 2.73 | 122.54 | (Liu et al., 2016) |
| | | 16.64 ± 8.61 | 0.96 ± 0.37 | 57.69 | |
| | | 14.87 ± 9.27 | 0.73 ± 0.22 | 49.09 | |
| Nanjing | 2013 | 52.3 ± 35.7 | 0.98 ± 0.35 | 18.72 | (Li et al., 2016) |
| | | 41.4 ± 27.2 | 1.10 ± 0.39 | 26.57 | |
| Shanghai | 2014 | 19.5 ± 9.98 | 1.85 | 94.87 | (Ming et al., 2017) |
| | | 16.5 ± 7.7 | 1.89 ± 0.72 | 114.73 | |
| Henan | 2018 | 22.5 ± 10.1 | 4.14 ± 1.57 | 183.91 | (Dong et |

| | | | | | |
|---|---|---|---|---|---|
| | | 3.2 ± 1.5 | 0.79 ± 0.32 | 246.53 | al., 2020) |
| Los Angeles | 2005-2018 | | 0.19 | | (Farahani et al., 2021) |
| | | | 0.014 | | |
| Thailand | 2019 | 8.02 ± 1.96 | 0.64 ± 0.09 | 79.80 | (Kayee et al., 2020) |
| Kaohsiung Harbor in Philippines | 2019 | 6.8 ± 1.53 | 0.53 ± 0.08 | 77.94 | (Tseng et al., 2021) |
| Manila Harbor in Taiwan | | 16.6 ± 6.6 | 1.30 ± 0.29 | 78.35 | |

4 Finally, throughout the paper it should be clarified that all particle concentrations reported are PM2.5.

We added the sentences in the Introduction part to clarified that all particle concentrations reported are $PM_{2.5}$ including total mass concentration of transition metals, water soluble ions and so on. And the mass concentration of $PM_{2.5}$ reported in the revised MS does not including particle water.

"…concentrations and the aerosol liquid water content (ALWC) on the aqueous reactant levels and the sulfate formation rate. All particle concentrations reported are fine particle matters with diameter below 2.5 μm ($PM_{2.5}$)."

Specific Comments

5 Abstract line 11, change hindes to hinders.

We change the word "hindes" to "hinders" in the revised MS abstract as "Lacking of detailed and comprehensive field data hinders the accurate evaluation of relative roles of prevailing sulfate formation pathways."

6 Line 83, can you provide plots of Fe and Mn vs PM2.5 mass? This would also help address one of the major points raised above?

We changed the Figure S4 in the revised SI. Total Mn mass concentration were estimated at the ratio of Fe/Mn equals to 28, thus the trend of Mn mass concentration was omitted in the figure.

[Figure]

Fig. S4. Time series of observed gas-phase pollutants concentrations, RH, Temperature, PM$_{2.5}$ mass loading, Fe as well as Cu total mass concentrations and calculated aerosol pH and water content and sulfate formation rates in these four haze periods in PKU-17 field campaign.

7 Table 1 Description (above the table) is incomplete. There are also aerosol particle metal species data but not noted, etc. Is the aerosol data PM2.5, please specify?

8 Table 1. Why not add the PM2.5 mass ranges for each pollution level to the table? Or put in the Table caption.

We added the aerosol particle metal species data notes in the title of Table and clarified the meaning of reported aerosol data in the title and notes of Table 1 in the revised MS. We also added the data of PM$_{2.5}$ dry mass concentration in Table 1. Please see the response above.

9 Lines 99-100. Is it really true that the main source (on a mass basis) for Fe, Cu and Mn is combustion? Is there a reference? Seems like road dust/tire and brake wear would be important as well as mineral dust. A more comprehensive assessment of the source of PM2.5 TMI in this study would also be useful given the conclusions (lines 277-285). Is coal fly ash really the main source for PM2.5 TMI in this region?

The issue of source apportionment analysis of aerosol metal in urban Beijing has been studied extensively. Based on these studies, the aerosol metal such as Fe are mainly crust related (Duan et al., 2012) and the peak concentrations of aerosol Fe and Mn reflected dust pollution caused by vehicle driving in traffic rush hours (Zhao et al., 2021). Aerosol Fe also shows a diurnal variation pattern that is high during the day and low at night, the distribution may be due to the intensive anthropogenic activities as

well as the driving of vehicles tire and brake wear in daytime causing elements from surface dust source entering PM$_{2.5}$ (Zhao et al., 2021). Cu and Mn are mainly from non-exhaust emissions of vehicles, fossil fuel combustion or metallurgy (Alexander et al., 2009; Duan et al., 2012; Zhao et al., 2021). Cu and Mn shows no seasonal pattern based on the studies of Zhao et al. (2021) while refers to Duan et al. (2012), higher concentration of Cu in winter indicating sources of coal combustion for heating in Beijing urban. Other studies also pointed out that combustion is an important source of aerosol Cu (Alexander et al., 2009; Schleicher et al., 2011). In the revised MS, we added other sources in addition to the combustion as "Atmospheric anthropogenic sources of transition metals such as iron (Fe) are mainly crust related and the peak concentration of Fe in Beijing is correlated to the vehicle driving in traffic rush hours. Copper (Cu), and manganese (Mn) are mainly from non-exhaust emissions of vehicles, fossil fuel combustion or metallurgy (Alexander et al., 2009; Duan et al., 2012; Zhao et al., 2021)."

On line 277-285, we changed the conclusion as "Compared to the gas-phase oxidants, the control of anthropogenic emissions of aerosol TMI is conducive to the reduction of secondary sulfates. The promotion of clean energy strategies aiming at reducing coal burning and vehicle emissions to improve air quality in North China has reduced not only the primary emissions of SO$_2$ but also the anthropogenic emissions of aerosol TMIs (Liu et al., 2018) and thus the production of secondary sulfate. What' more, China's ecological and environmental protection measures for tree planting and afforestation are conducive to reducing the generation of dust especially in the spring can further reducing the quality of metal Fe concentrations in aerosols."

10 What are the units for data in table S9?

The unit of metal concentration is ng/m$^3$ in Table S9. We added the unit is the revised Supplementary Information:

Table S9. Concentration of transition metals in PM$_{2.5}$ in urban areas.

| Sampling site | Period | Method | Fe (ng/m$^3$) | Mn (ng/m$^3$) | Cu (ng/m$^3$) | References |
|---|---|---|---|---|---|---|
| China, Beijing, Urban | 2018.8-2019.8 | XRF | 596 | 27.9 | 7.37 | Zhao et al. (2021) |
| China, Beijing, Urban | 2015.9-2016.1 | XRF | 686 | 60.2 | 25.1 | Zhang et al. (2019) |
| China, Beijing, Urban | 2016.6-2017.5 | ED-XRF | 738 | 37 | 32 | Cui et al. (2019) |
| China, Beijing, Urban | 2014.1-10 | ICP-AES | 1650 | 55 | 108 | Gao et al. (2018) |
| China, Beijing, Urban | 2016.1-2017.5 | XRF | 629 | 32 | 24 | Cui et al. (2020) |
| China, Beijing, Urban | 2016.1 | ICP-AES | 2823 | 92.3 | 48 | Duan et al. (2012) |
| China, | 2017.10- | XRF | 1361 | 157 | 29.2 | He et al. (2019) |

| | | | | | | |
|---|---|---|---|---|---|---|
| Zhengzhou, Urban | 2018.7 | | | | | |
| China, Nanjing, Urban | 2016.12-2017.12 | XRF | 577 | 48.9 | 27.2 | Yu et al. (2019) |
| China, Shanghai, Urban | 2016.3-2017.2 | ED-XRF | 410 | 32 | 12 | Chang et al. (2017) |
| Canada, Hamilton, Urban | 2014.1-2017.6 | XRF | 49.6 | 0.83 | 2.76 | Sofowote et al. (2019) |
| India, New Delhi, Urban | 2013.1-2016.12 | WD-XRF | 780 | 10 | 100 | Jain et al. (2020) |

11 Fig 1 caption needs work; does plot (c) really show diurnal trends, keep same scale for SO4 in (c) and (d), and define SOR and indicate it is the line in plots (e and (f)).

Figure 1 shows the three-hour averaged sulfate formation rates during haze periods in the scale of 6 days. We added the modeled $SO_4^{2-}$ concentration in summer haze periods in panel (d) in the revised MS as shown in the following response. In the section 2.2 the second paragraph, SOR is defined as the ratio of mole concentration of $SO_2$ to the sum of $SO_2$ and $SO_4^{2-}$.

"Compared to the total Fe concentration, it is more effective to evaluate the impact of αFe (III) on sulfate formation. The relationship between αFe (III) and SOR ($\equiv n(SO_2)/n(SO_2+SO_4^{2-})$), defined as the ratio of mole concentration of $SO_2$ with the sum of $SO_2$ and $SO_4^{2-}$ mole concentrations) in…"

And we changed the Figure 1 in the revised MS as:

[Figure]

**Figure 1. Three-hour average sulfate formation rates during haze periods in winter and summer (a)&(b), corresponding effective Fe (III) concentrations and sulfate concentrations (c)&(d), sulfate formation rates (the histogram) and SOR (the** dotted lines**) in different pollution levels in two field campaigns (e)&(f).**

12 Line 150-151, correlation is not causation, reword to say the correlation is consistent with…

We revised the incorrect statement about the relevant in the revised MS section 2.2 as "Obvious correlations between αFe (III) and sulfate concentration shown in **Fig. 1 (c) and (d)** were observed in the haze periods both in summer ($R^2$=0.63) and winter ($R^2$=0.71) and the correlation is consistent with the important contributions from aqTMI pathway to the sulfate formation."

13 Line 153, what does n(SO2) mean?

The definition of $n(SO_2)$ in the MS is the mole concentration of $SO_2$, and $n(SO_2+SO_4^{2-})$ is defined as the sum of mole concentration of $SO_2$ and $SO_4^{2-}$. We added this piece of definition in the revised MS in section 2.2 as "The relationship between αFe (III) and SOR ($\equiv n(SO_2)/n(SO_2+SO_4^{2-})$), defined as the ratio of mole concentration of $SO_2$ with

the sum of SO$_2$ and SO$_4^{2-}$ mole concentrations) in PKU-17 winter field campaign was shown in **SI Figure S5**"

14 It should be stated that Eq(1) is simply the conversion of sulfate formation rate in the aerosol water (ie, per mL water) to sulfate formation per m3 of air. There is nothing special about this.

That's true about the comment on Equation (1). Anyway, using different units to look at the rate of sulfate formation is of vital importance to the study of the formation of secondary sulfate aerosols, which can help us better think about the proportion of the contributions from different pathways in different chemical regime. In the revised MS, we added the sentences above the Equation (1) as "In the calculation, we changed the unit of sulfate formation rate from µg/m$^3_{air}$ to mol/s·L$_{water}$ and the sulfate formation rate can be calculated via the following equation with the modeled $\frac{dS(VI)}{dt}$ ($M/s$ ):…"

15 Line 186-187 and on is not clear. Is the point that the equilibrium amount of H2O2, O3, and NO2 in units of mass/m3 air is controlled by the amount of ALW, ie there is equilibrium between gas and particle water for these oxidants formed in the gas phase. Is the idea that TMI is a primary aerosol (that is not likely really true, it may be true for the total elements, Fe, or Mn, but not the ions) so does not depend on ALW? So the idea is that ALW does not affect TMI levels in solution by affecting the solubility of the overall metal form of the specific species (ie, fig 3 shows insensitivity of pH to ALW, which has been pointed out in other papers (eg, Wong,et al., 2020, Env Sci Tech, 54: 7088-96.)

It was accurately the meaning of line 186-187 and we changed the sentences below the Equation (1) in the revised MS and added the reference to explain the irrelevance of aerosol pH with ALWC.

"The equilibrium amount of H$_2$O$_2$, O$_3$, and NO$_2$ in units of µg/m$^3_{air}$ is controlled by the amount of ALW, ie there is equilibrium between gas and particle water for these oxidants formed in the gas phase. And total amount of metal elements, Fe, Cu or Mn is not dependent on aerosol water content. Aerosol water content does not affect TMI levels in solution by affecting the solubility of the overall metal form of the specific species (**Fig.3** shows insensitivity of pH to ALWC, which has been pointed out in other papers (Wong et al., 2020)."

16 What does PM$_{2.5}$ represent in Fig 3, the total mass including particle water?

The reported PM$_{2.5}$ mass concentration does not include particle water in the original and revised MS. The mass concentration of PM$_{2.5}$ was measured by commercial Ambient Particulate Monitor (TEOM). We added this sentence in the revised MS in Section Methods 1.

17 Line 199-200. What does transition metal mass will not increase mean? The mass concentration of TMI in air or the liquid concentration? Care must be taken in this whole section on what concentration (in air or in ALW) is being discussed.

This part mainly discusses the influence of ALWC on the sulfate formation rates from Mn-surface and aqTMI pathways. Aqueous TMI mole concentration will not increase with the aerosol hygroscopic growth. With the aerosol hygroscopic growth, the increasing of transition metal total mass in air is slower than aerosol water mass in PKU-17. The ratio of Fe total mass with ALWC decreasing with $PM_{2.5}$ mass shown in Fig. S7 indicating a "dilution effect" which means aqueous mole concentration of TMI decreasing with higher aerosol water content. We added the above discussion in Section 2.3 penultimate paragraph in the revised MS as "Due to the obvious heterogeneous reactions contribution to sulfate formation in winter, we evaluated the influence of ALWC on sulfate formation pathways in winter. TMI relevant pathways including aqTMI and Mn-surface pathway were dominate in all range of ALWC as illustrated in **Fig.3**. In PKU-17 field campaign, with the increasing of ALWC from 1 to 150 $\mu g/m^3$, the ratio of Mn-surface/aqTMI continuously decreased mainly because of the decreasing particle specific surface areas. Mn-surface contributed most in lower ALWC range where particle specific surface area was high and provide more reaction positions. Aqueous transition metal ions mole concentration decreasing with the aerosol hygroscopic growth indicating a "dilution effect" as shown in **Fig. S7**. With the aerosol hygroscopic growth, the increasing of transition metal total mass in air is slower than water mass in PKU-17. The ratio of Fe total mass ($Fe_t$)/ALWC decreasing with $PM_{2.5}$ mass. Previous globle scale observations (Sholkovitz et al., 2012) of ~1100 samples also showed the hyperbolic trends of Fe solubility with total Fe mass. Higher activity coefficients and lower aqueous TMI concentration led to the emergence of "high platform" of the aqTMI pathways contribution to sulfate formation in the range of 50-150 $\mu g/m^3$ ALWC (ie, higher effective aqueous TMI in this range). While ALWC exceeding 150 $\mu g/m^3$ in winter, the increase of activity coefficients could not promote the rate of aqTMI. Due to the slight increase of aerosol pH and the dilution effect of aerosol hygroscopic growth on TMI when ALWC exceeding 150 $\mu g/m^3$ as discussed above, the importance of aqTMI and Mn-surface contributions were lowered. At this time, the contributions of external oxidizing substances pathways such as $H_2O_2$, $NO_2$ or $O_3$ may rise in the proper pH range as illustrated in **Fig.4**. In winter fog or cloud conditions with higher water content, the contribution from TMI may decrease a lot for their low solubility and concentrations."

18 Line 200-201. This is not clear and Fig S7 is not clear how it supports this idea of a dilution effect. Define Ft in Fig S7.
The meaning of "dilution effect" was explained in the above response. $Fe_t$ in Fig. S7 means the total mass concentration of $PM_{2.5}$ Fe in air. We changed the title of this figure as: Fig. S7. The "dilution effect" of Fe total mass concentration in air ($Fe_t$) and ALWC increasing with PM mass in winter and summer.

19 Line 203, it is not clear how the results of Sholkovitz apply here as they are looking at regions largely influenced by mineral dust and some combustion, here the authors state that the metals are from combustion. There is an inconsistency.

We changed the inaccurate statements about the source of aerosol metal including Fe, Cu and Mn in the revised MS, please refer to the above response. Aerosol Fe in Beijing urban area mainly related to the mineral dust and vehicle emissions.

20 Line 204, I do not understand the statement, the importance of aqTMI and Mn-surface contributions were lowered. Why is it lowered, pH in Fig 3 changes very little at ALWC ? 150 ug/m3. This whole section on the effect of water is very confusing. Can the authors give a physical explanation on what the effect of liquid water is on the ambient air concentration of transition metal ions in $PM_{2.5}$.

Aerosol liquid water content has tiny influence on the ambient air total mass transition metal in $PM_{2.5}$ while has the "dilution effect" as discussed above and influence the aqueous TMI concentrations. At the same time, the activity coefficient of TMI increase with aerosol hygroscopic growth led to the emergence of high platforms of the aqTMI pathways contribution to sulfate formation in the range of 50-150 $\mu g/m^3$ ALWC. With ALWC exceeding 150 $\mu g/m^3$, the effective aqueous TMI concentration (the product of TMI mole concentrations and activity coefficients) decreasing otherwise weaken the importance of aqTMI and Mn-surface. When considering the liquid-phase kinetic reaction to produce sulfate, we pay more attention to the change of the liquid-phase ion concentration in aerosol water rather than the change of the total concentration in the air. We reword this paragraph in the revised MS as mentioned in Response 17.

21 Fig 4 should have plots labeled (a) and (b)
We added the icons in the Figure 4 in the revised MS as follows:
**Figure 4. Bar graph showing modelled contributions of various pathways to sulfate formation under different pollution conditions.**

[Figure]

Different pollution conditions including clear ($PM_{2.5}$ smaller than 35 $\mu g/m^3$) in winter PKU 2017 (C_winter_4.3) and summer WD 2014 (C_summer_4.5); pollution ($PM_{2.5}$ larger than 75 $\mu g/m^3$) in PKU 2017 (H_winter_4.8), WD 2014 (H_summer_4.2); fog conditions used in a previous study(Xue et al., 2016) (Fog_winter_6.0) and cloud conditions (Cloud_5.5) simulated by Seinfeld and Pandis (2016). The number in each

label indicates the average pH value chosen in these calculations. We assumed that the cloud water content is 0.1 g/m$^3$ in the last condition, and reduced the H$_2$O$_2$ concentration to 0.1 ppb compared to the high value used before (Seinfeld and Pandis, 2016).

22 Line 253 to 255 is not clear (While as mentioned above,

In PKU-17 field campaign, with the increasing of ALWC from 1 to 150 μg/m$^3$, the ratio of Mn-surface/aqTMI continuously decreased mainly because of the decreasing particle specific surface areas as shown in Fig.3 panel (b) dotted lines. What's more, the organic coating of aerosol particles can largely reduce the reactivity of surface heterogeneous reactions (Zelenov et al., 2017; Anttila et al., 2006; Folkers et al., 2003; Ryder et al., 2015) and may cause the Mn-surface pathway less important. The surface reaction of SO$_2$ with Mn and other metals in actual aerosol conditions remain unclear, and the relevant calculation results of WD-14 and PKU-17 in this paper represent the upper limit of Mn-surface contribution. We added more references in the revised MS in order to explain the propose of this paragraph:

"While as mentioned above, the ratio of contributions from Mn-surface/aqTMI to produce sulfate will decrease with aerosol hygroscopic growth owning higher ALWC and lower specific surface areas (as shown in Fig.3 panel (b) black dotted line). What's more, the organic coating of aerosol particles can largely reduce the reactivity of surface heterogeneous reactions (Zelenov et al., 2017; Anttila et al., 2006; Folkers et al., 2003; Ryder et al., 2015) and may cause the Mn-surface pathway less important."

23 Line 260-261 reword, not clear.
We reword this part of discussion in the revised MS as follows.
"The organic coating can effectively reduce the reactive sites in the surface of particles hence reduce the reaction probability of SO$_2$ with surface metal. In the other hand, the widespread presence of aerosol organic coating can also influence the bulk SO$_2$ catalysed by aqueous TMI but not only the surface reactions. This effect is mainly achieved by the change of SO$_2$ solubility and diffusion coefficient rather than the rates of catalytic reactions with TMI. Although the solubility of SO$_2$ in organic solvent changes a lot with the component of organic (Zhang et al., 2013; Huang et al., 2014), according to previous studies of SO$_2$ uptake coefficient with sea-salt aerosol (Gebel et al., 2000) and secondary organic aerosol (SOA) (Yao et al., 2019), no obvious uptake coefficient reduction was observed with the organic coating further proving the minor influence of the organic coating on bulk reaction rates. The catalytic reaction of SO$_2$ with aqTMI may less affected by aerosol organic coating compared to SO$_2$ with Mn-surface. For these reasons, the surface reaction of SO$_2$ with Mn and other metals in actual aerosol conditions remain unclear with high uncertainties and need further evaluation. The relevant calculation results of WD-14 and PKU-17 in this paper represent the upper limit of Mn-surface contribution. The missing contribution in WD-14 polluted conditions may mainly come from organic photosensitizing molecules such as HULIS (Wang et al., 2020) under stronger UV in summer or other SOA coupled

mechanisms."

24 Line 324, state-state?
We deleted the incorrect wording "state-state" and changed the sentence as "…the PKU-MARK model produced the concentrations of aqueous reactants in one-hour resolution including…" in the revised MS section 4.2.

**Reference:**

Alexander, B., Park, R. J., Jacob, D. J., and Gong, S.: Transition metal-catalyzed oxidation of atmospheric sulfur: Global implications for the sulfur budget, Journal of Geophysical Research: Atmospheres, 114, https://doi.org/10.1029/2008JD010486, 2009.

Anttila, T., Kiendler-Scharr, A., Tillmann, R., and Mentel, T. F.: On the Reactive Uptake of Gaseous Compounds by Organic-Coated Aqueous Aerosols:   Theoretical Analysis and Application to the Heterogeneous Hydrolysis of N2O5, The Journal of Physical Chemistry A, 110, 10435-10443, 10.1021/jp062403c, 2006.

Chang, Y., Huang, K., Deng, C., Zou, Z., Liu, S., and Zhang, Y.: First long-term and near real-time measurement of atmospheric trace elements in Shanghai, China, Atmos. Chem. Phys. Discuss., in Review, 2017.

Cui, Y., Ji, D., He, J., Kong, S., and Wang, Y.: In situ continuous observation of hourly elements in PM2. 5 in urban beijing, China: Occurrence levels, temporal variation, potential source regions and health risks, Atmos. Environ., 222, 117164, 2020.

Cui, Y., Ji, D., Chen, H., Gao, M., Maenhaut, W., He, J., and Wang, Y.: Characteristics and sources of hourly trace elements in airborne fine particles in urban Beijing, China, Journal of Geophysical Research: Atmospheres, 124, 11595-11613, 2019.

Dong, Z., Su, F., Zhang, Z., and Wang, S.: Observation of chemical components of PM2.5 and secondary inorganic aerosol formation during haze and sandy haze days in Zhengzhou, China, Journal of Environmental Sciences, 88, 316-325, https://doi.org/10.1016/j.jes.2019.09.016, 2020.

Duan, J. C., Tan, J. H., Wang, S. L., Hao, J. M., and Chail, F. H.: Size distributions and sources of elements in particulate matter at curbside, urban and rural sites in Beijing, Journal of Environmental Sciences, 24, 87-94, 10.1016/s1001-0742(11)60731-6, 2012.

Farahani, V. J., Soleimanian, E., Pirhadi, M., and Sioutas, C.: Long-term trends in concentrations and sources of PM2.5–bound metals and elements in central Los Angeles, Atmos. Environ., 253, 118361, https://doi.org/10.1016/j.atmosenv.2021.118361, 2021.

Folkers, M., Mentel, T. F., and Wahner, A.: Influence of an organic coating on the reactivity of aqueous aerosols probed by the heterogeneous hydrolysis of N2O5, Geophysical Research Letters, 30, https://doi.org/10.1029/2003GL017168, 2003.

Gao, J., Wang, K., Wang, Y., Liu, S., Zhu, C., Hao, J., Liu, H., Hua, S., and Tian, H.: Temporal-spatial characteristics and source apportionment of PM2. 5 as well as its associated chemical species in the Beijing-Tianjin-Hebei region of China, Environ. Pollut., 233, 714-724, 2018.

Gebel, M. E., Finlayson-Pitts, B. J., and Ganske, J. A.: The uptake of SO2 on synthetic sea salt and

some of its components, Geophysical Research Letters, 27, 887-890, https://doi.org/10.1029/1999GL011152, 2000.

He, R.-D., Zhang, Y.-S., Chen, Y.-Y., Jin, M.-J., Han, S.-J., Zhao, J.-S., Zhang, R.-Q., and Yan, Q.-S.: Heavy metal pollution characteristics and ecological and health risk assessment of atmospheric PM2. 5 in a living area of Zhengzhou City, Huan jing ke xue= Huanjing kexue, 40, 4774-4782, 2019.

Ho, K. F., Lee, S. C., Chan, C. K., Yu, J. C., Chow, J. C., and Yao, X. H.: Characterization of chemical species in PM2.5 and PM10 aerosols in Hong Kong, Atmos. Environ., 37, 31-39, https://doi.org/10.1016/S1352-2310(02)00804-X, 2003.

Huang, K., Xia, S., Zhang, X.-M., Chen, Y.-L., Wu, Y.-T., and Hu, X.-B.: Comparative Study of the Solubilities of SO2 in Five Low Volatile Organic Solvents (Sulfolane, Ethylene Glycol, Propylene Carbonate, N-Methylimidazole, and N-Methylpyrrolidone), Journal of Chemical & Engineering Data, 59, 1202-1212, 10.1021/je4007713, 2014.

Ibusuki, T. and Takeuchi, K.: Sulfur dioxide oxidation by oxygen catalyzed by mixtures of manganese(II) and iron(III) in aqueous solutions at environmental reaction conditions, Atmospheric Environment (1967), 21, 1555-1560, https://doi.org/10.1016/0004-6981(87)90317-9, 1987.

Jain, S., Sharma, S., Vijayan, N., and Mandal, T.: Seasonal characteristics of aerosols (PM2. 5 and PM10) and their source apportionment using PMF: a four year study over Delhi, India, Environ. Pollut., 262, 114337, 2020.

Kayee, J., Sompongchaiyakul, P., Sanwlani, N., Bureekul, S., Wang, X., and Das, R.: Metal Concentrations and Source Apportionment of PM2.5 in Chiang Rai and Bangkok, Thailand during a Biomass Burning Season, ACS Earth and Space Chemistry, 4, 1213-1226, 10.1021/acsearthspacechem.0c00140, 2020.

Lai, S., Zhao, Y., Ding, A., Zhang, Y., Song, T., Zheng, J., Ho, K. F., Lee, S.-c., and Zhong, L.: Characterization of PM2.5 and the major chemical components during a 1-year campaign in rural Guangzhou, Southern China, Atmos. Res., 167, 208-215, https://doi.org/10.1016/j.atmosres.2015.08.007, 2016.

Li, H. M., Wang, Q. G., Yang, M., Li, F. Y., Wang, J. H., Sun, Y. X., Wang, C., Wu, H. F., and Qian, X.: Chemical characterization and source apportionment of PM2.5 aerosols in a megacity of Southeast China, Atmos. Res., 181, 288-299, 10.1016/j.atmosres.2016.07.005, 2016.

Liu, B., Song, N., Dai, Q., Mei, R., Sui, B., Bi, X., and Feng, Y.: Chemical composition and source apportionment of ambient PM2.5 during the non-heating period in Taian, China, Atmos. Res., 170, 23-33, https://doi.org/10.1016/j.atmosres.2015.11.002, 2016.

Liu, J., Chen, Y., Chao, S., Cao, H., Zhang, A., and Yang, Y.: Emission control priority of $PM_{2.5}$-bound heavy metals in different seasons: A comprehensive analysis from health risk perspective, Sci. Total Environ., 644, 20-30, https://doi.org/10.1016/j.scitotenv.2018.06.226, 2018.

Martin, L. R. and Hill, M. W.: The iron catalyzed oxidation of sulfur: Reconciliation of the literature rates, Atmospheric Environment (1967), 21, 1487-1490, 1967.

Martin, L. R. and Hill, M. W.: The effect of ionic strength on the manganese catalyzed oxidation of sulfur (IV), Atmospheric Environment (1967), 21, 2267-2270, 1987.

Ming, L. L., Jin, L., Li, J., Fu, P. Q., Yang, W. Y., Liu, D., Zhang, G., Wang, Z. F., and Li, X. D.: PM2.5 in the Yangtze River Delta, China: Chemical compositions, seasonal variations, and regional pollution events, Environ. Pollut., 223, 200-212, 10.1016/j.envpol.2017.01.013, 2017.

Paris, R. and Desboeufs, K. V.: Effect of atmospheric organic complexation on iron-bearing dust

solubility, Atmos. Chem. Phys., 13, 4895-4905, 10.5194/acp-13-4895-2013, 2013.

Ryder, O. S., Campbell, N. R., Morris, H., Forestieri, S., Ruppel, M. J., Cappa, C., Tivanski, A., Prather, K., and Bertram, T. H.: Role of Organic Coatings in Regulating N2O5 Reactive Uptake to Sea Spray Aerosol, The Journal of Physical Chemistry A, 119, 11683-11692, 10.1021/acs.jpca.5b08892, 2015.

Schleicher, N. J., Norra, S., Chai, F., Chen, Y., Wang, S., Cen, K., Yu, Y., and Stüben, D.: Temporal variability of trace metal mobility of urban particulate matter from Beijing – A contribution to health impact assessments of aerosols, Atmos. Environ., 45, 7248-7265, https://doi.org/10.1016/j.atmosenv.2011.08.067, 2011.

Seinfeld, J. H. and Pandis, S. N.: Atmospheric chemistry and physics: from air pollution to climate change, John Wiley & Sons2016.

Sholkovitz, E. R., Sedwick, P. N., Church, T. M., Baker, A. R., and Powell, C. F.: Fractional solubility of aerosol iron: Synthesis of a global-scale data set, Geochimica et Cosmochimica Acta, 89, 173-189, https://doi.org/10.1016/j.gca.2012.04.022, 2012.

Sofowote, U. M., Di Federico, L. M., Healy, R. M., Debosz, J., Su, Y., Wang, J., and Munoz, A.: Heavy metals in the near-road environment: Results of semi-continuous monitoring of ambient particulate matter in the greater Toronto and Hamilton area, Atmospheric Environment: X, 1, 100005, 2019.

Tapparo, A., Di Marco, V., Badocco, D., D'Aronco, S., Soldà, L., Pastore, P., Mahon, B. M., Kalberer, M., and Giorio, C.: Formation of metal-organic ligand complexes affects solubility of metals in airborne particles at an urban site in the Po valley, Chemosphere, 241, 125025, https://doi.org/10.1016/j.chemosphere.2019.125025, 2020.

Tseng, Y.-L., Wu, C.-H., Yuan, C.-S., Bagtasa, G., Yen, P.-H., and Cheng, P.-H.: Inter-comparison of chemical characteristics and source apportionment of PM2.5 at two harbors in the Philippines and Taiwan, Sci. Total Environ., 793, 148574, https://doi.org/10.1016/j.scitotenv.2021.148574, 2021.

Wang, X., Gemayel, R., Hayeck, N., Perrier, S., Charbonnel, N., Xu, C., Chen, H., Zhu, C., Zhang, L., Wang, L., Nizkorodov, S. A., Wang, X., Wang, Z., Wang, T., Mellouki, A., Riva, M., Chen, J., and George, C.: Atmospheric Photosensitization: A New Pathway for Sulfate Formation, Environ. Sci. Technol., 54, 3114-3120, 10.1021/acs.est.9b06347, 2020.

Wang, Y. N., Jia, C. H., Tao, J., Zhang, L. M., Liang, X. X., Ma, J. M., Gao, H., Huang, T., and Zhang, K.: Chemical characterization and source apportionment of PM2.5 in a semi-arid and petrochemical-industrialized city, Northwest China, Sci. Total Environ., 573, 1031-1040, 10.1016/j.scitotenv.2016.08.179, 2016.

Wong, J. P. S., Yang, Y., Fang, T., Mulholland, J. A., Russell, A. G., Ebelt, S., Nenes, A., and Weber, R. J.: Fine Particle Iron in Soils and Road Dust Is Modulated by Coal-Fired Power Plant Sulfur, Environ. Sci. Technol., 54, 7088-7096, 10.1021/acs.est.0c00483, 2020.

Wozniak, A. S., Shelley, R. U., McElhenie, S. D., Landing, W. M., and Hatcher, P. G.: Aerosol water soluble organic matter characteristics over the North Atlantic Ocean: Implications for iron-binding ligands and iron solubility, Marine Chemistry, 173, 162-172, https://doi.org/10.1016/j.marchem.2014.11.002, 2015.

Xue, J., Yuan, Z., Griffith, S. M., Yu, X., Lau, A. K. H., and Yu, J. Z.: Sulfate Formation Enhanced by a Cocktail of High NOx, SO2, Particulate Matter, and Droplet pH during Haze-Fog Events in Megacities in China: An Observation-Based Modeling Investigation, Environ. Sci. Technol., 50, 7325-7334, 10.1021/acs.est.6b00768, 2016.

Yao, M., Zhao, Y., Hu, M. H., Huang, D. D., Wang, Y. C., Yu, J. Z., and Yan, N. Q.: Multiphase

Reactions between Secondary Organic Aerosol and Sulfur Dioxide: Kinetics and Contributions to Sulfate Formation and Aerosol Aging, Environmental Science & Technology Letters, 6, 768-774, 10.1021/acs.estlett.9b00657, 2019.

Yin, L., Niu, Z., Chen, X., Chen, J., Xu, L., and Zhang, F.: Chemical compositions of PM2.5 aerosol during haze periods in the mountainous city of Yong'an, China, Journal of Environmental Sciences, 24, 1225-1233, https://doi.org/10.1016/S1001-0742(11)60940-6, 2012.

Yu, Y., He, S., Wu, X., Zhang, C., Yao, Y., Liao, H., Wang, Q. g., and Xie, M.: PM2. 5 elements at an urban site in Yangtze River Delta, China: High time-resolved measurement and the application in source apportionment, Environ. Pollut., 253, 1089-1099, 2019.

Zelenov, V. V., Aparina, E. V., Kashtanov, S. A., and Shardakova, E. V.: Kinetics of NO3 uptake on a methane soot coating, Russian Journal of Physical Chemistry B, 11, 180-188, 10.1134/s1990793117010146, 2017.

Zhang, B., Zhou, T., Liu, Y., Yan, C., Li, X., Yu, J., Wang, S., Liu, B., and Zheng, M.: Comparison of water-soluble inorganic ions and trace metals in $PM_{2.5}$ between online and offline measurements in Beijing during winter, Atmos. Pollut. Res., 10, 1755-1765, https://doi.org/10.1016/j.apr.2019.07.007, 2019.

Zhang, N., Zhang, J., Zhang, Y., Bai, J., and Wei, X.: Solubility and Henry's law constant of sulfur dioxide in aqueous polyethylene glycol 300 solution at different temperatures and pressures, Fluid Phase Equilibria, 348, 9-16, https://doi.org/10.1016/j.fluid.2013.03.006, 2013.

Zhao, S., Tian, H., Luo, L., Liu, H., Wu, B., Liu, S., Bai, X., Liu, W., Liu, X., Wu, Y., Lin, S., Guo, Z., Lv, Y., and Xue, Y.: Temporal variation characteristics and source apportionment of metal elements in PM2.5 in urban Beijing during 2018–2019, Environ. Pollut., 268, 115856, https://doi.org/10.1016/j.envpol.2020.115856, 2021.

---

## Author Comment (AC2)

**Response to the comments of referee #2**

The manuscript entitled "A Comprehensive Observational Based Multiphase Chemical Model Analysis of the Sulfur Dioxide Oxidations in both Summer and Winter" by Song et al. presents the comprehensive evaluation of the contribution of different sulfate production pathways in both summer and winter by their self-developed multiphase box model (PKU-MARK). The model includes nearly all the established sulfate production pathways, providing valuable insights into the sulfate formation evaluation. Overall, I have some concerns that need the authors to clarify before that I can recommend publication in Atmospheric Chemistry and Physics.

We thank the reviewer for the helpful comments. The referee's comments are first given in black type, followed by our response to each in turn in blue type. Any changes to the manuscript in response to the comments are then given in quotation marks in red type.

1 The concentration of TMIs is vital in this work since the two dominant sulfate production pathways the authors proposed are aqTMI and Mn-surface. An online monitor measured the Fe and Cu concentrations. Due to the lack of Mn data, the authors propose a fixed ratio of Fe/Mn to stimulate the concentration of Mn. So how about the uncertainty of this method? It is better to compare the concentration of Mn with literature results in the same region.

Under the assumption of fixed Fe/Mn air  $PM_{2.5}$  mass concentration ratio as 28, the air mass concentrations of Mn were calculated and shown in Table 1. Mn averaged mass concentration range from 12.4±9.4 ng/m3 in clean situation to 46.5±10.3 ng/m3 in highly polluted situation in PKU-17 observation. Compared to measurement results of Mn in the same region reviewed in the Table S9 (27.9 - 92.3 ng/m3), values were slightly lower. While in the measurements of Cui et al. (2019) in 2016.6 to 2017.5 which is the closest measurement time, averaged Mn concentration was  $32\pm25$  ng/m3 in non-heating periods and  $35\pm36$  ng/m3 in heating periods. The results were in consist with our estimation. However, fixed ratio of Fe/Mn leads to uncertainties of effective aqueous TMI concentration when evaluating the sulfate formation. We have revised the MS to supplement the sensitivity analysis on the concentration of aqueous TMI. Please refer to the response below.

2 The authors state that the average soluble percentage of Fe and Mn in winter polluted conditions was 0.79% and 19.83%. However, the water-soluble fraction of Fe and Mn may change a lot in different regions, as stated in the manuscript. Also, the solubility may change under clean conditions and polluted conditions. It is better to add some discussion about the sensitivity of the solubility of Fe and Mn to the model results. We added the following discussion of transition metal sensitivity on sulfate formation in PKU-17 winter field campaign in the revised SI Text S4, Figure S9 and Table S10. Water soluble fraction of Fe, Mn and Cu can vary over a large range. A large part of the

soluble metals is in the form of soluble organic complexes or hydroxides rather than ions in aerosol particles. There are evidences that the existence of various aerosol water soluble organic acids (oxalate, malonate, tartrate and humic acid) cause an enhancement of Fe, Cu and Mn solubility and the formation of metal-organic complex (Paris and Desboeufs, 2013; Wozniak et al., 2015; Tapparo et al., 2020). What's more, the dissolution of Fe and Mn is highly influenced by aerosol pH. Circumneutral pH leads to a supersaturated soluble Fe (III), which then precipitates out of the solution. For these reasons, the promotion of metal solubility may have non-proportional influences on the aqueous concentration TMI. We conducted the sensitivity analysis for the solubility of Fe from 1% to 15% (Scenario one with fixed aqueous Mn and Cu concentration consist with the base run in the MS, Scenario two with fixed ratio of soluble Fe/Mn and Fe/Cu mass, ie, Mn solubility in the range of 10% to 100%, Cu in 5% to 75%, as shown in Table S10). Other aerosol component concentration, ionic strength, ALWC, observed meteorological parameters and trace gases concentrations stay consistent with the base run.

Figure S.9 Sensitivity analysis of transition metal including Fe, Mn and Cu solubility influences on the averaged sulfate formation rates in PKU-17 field observation. Dotted lines in the figure show the cluster averaged results with a pH span of 0.5 under actual ambient conditions with different transition metal solubilities.

Table S10. Base run and scenarios of the solubility sensitivity analysis.

| Sensitivity | Solubility of transition metals | Sulfate formation contribution in                    |  |
|-------------|---------------------------------|------------------------------------------------------|--|
| Analysis    |                                 | haze pH range (4.2-5.2) ( $\mu$ g/m 3 /h) |  |
| Base Run    | 5% Fe + 50% Mn + 25% Cu         | 0.80 - 2.58                                          |  |
| Scenario 1  | 15% Fe + 50% Mn + 25% Cu        | 3.49 - 8.57                                          |  |
|             | 1% Fe + 50% Mn + 25% Cu         | 0.05 - 0.16                                          |  |
| Scenario 2  | 15% Fe + 100% Mn + 75% Cu       | 12.97 - 32.87                                        |  |
|             | 1% Fe + 10% Mn + 5% Cu          | 0.009 - 0.004                                        |  |

As shown in Figure S.9, In the range of winter haze periods pH (4.2-5.2), averaged sulfate formation rates in PKU-17 field observation is non-proportional to the initial transition metal solubility. Fe solubility increasing from 1% to 5% will cause d[S(VI)]/dt to increase over an order of magnitude, and increasing to 15% cause no obvious effect when pH smaller than 4.2, while obvious effect the pH ranging from 4.2 to 6. This phenomenon may be due to the piecewise calculation equations of TMIcatalysis oxidizing SO2 as mentioned in the SI and following. In the presence of TMI organic complexes and redox reactions, this equation may need to be further verified, but verification is not within the scope of this study. It is obvious that the d[S(VI)]/dt changes caused by the proportional expansion of the solubility of the three transition metals (Scenario 2) is more significant especially when the solubility is reduced to 1%+10%+5%. Increasing of solubility to 15%Fe+100%Mn+75%Cu can increase sulfate formation rate to 5-84 times higher than in base run during haze periods pH as 4.2-5.2. This can explain to a certain extent that excessive TMI concentrations will not cause a sharp increase in d[S(VI)]/dt, which may be due to the buffering effect caused by the formation of organic complexes.

Part of Table S2. Aqueous-phase reaction rate expressions, rate constants (k) and influence of ionic strength (Is) on k for sulfate production in aerosol particle condensed phase.

| Oxidant                                              | The reaction rate expressions $(R_{S(IV)+oxi})$ ,                                   | Notes              | References             |  |
|------------------------------------------------------|-------------------------------------------------------------------------------------|--------------------|------------------------|--|
| S                                                    | constants (k) and influence of $I_s$ (in unit of                                    |                    |                        |  |
| M) on k a                                 |                                                                                     |                    |                        |  |
| TMI+O 2 ±                      | $k_{6}[H^{+}]^{-0.74}[S(IV)][Mn(II)][Fe(III)] (pH \leq$                             |                    | Ibusuki and Takeuchi   |  |
|                                                      | 4.2)                                                                                |                    | (1987)                 |  |
|                                                      | $k_6 = 3.72 \times 10^7 \times e^{(-8431.6 \times (1/T - 1/297))} M^{-2} s^{-1}$    |                    |                        |  |
| $k_7[H^+]^{0.67}[S(IV)][Mn(II)][Fe(III)] (pH > 4.2)$ |                                                                                     |                    |                        |  |
|                                                      | $k_7 = 2.51 \times 10^{13} \times e^{(-8431.6 \times (1/T - 1/297))} M^{-2} s^{-1}$ |                    |                        |  |
|                                                      | $b_{4}\sqrt{I_{s}}$                                                                 | $I_{\rm s, max} =$ | Martin and Hill (1987, |  |
|                                                      | $\log_{10}\left(\frac{1}{k^{I_{s}=0}}\right) = \frac{1}{1+\sqrt{I_{s}}}$            | 2 M                | 1967)                  |  |
|                                                      | $b_4$ is in range of $-4$ to $-2$                                                   |                    |                        |  |

3 The authors declare that their result is consistent with the result of the WRF-CHEM study. However, in the cited work, the ionic strength inhibition effect was not included. More discussion about the results is needed.

In the latest WRF-CHEM study of Tao et al. (2020), the concentrations of Fe and Mn were modeled as the minimum of the solubility of metals regardless of the acidity of aerosol water and ion equilibrium depending on pH. The ionic strength inhibition effect was not included. Using the same Fe/Mn concentration calculation method and considering the ionic strength, Wang et al. (2021) pointed out in the latest research results that aqTMI catalysis only accounts for less than 1% of sulfate formation. Our rough analysis suggests that the calculation method of Fe/Mn may underestimate the actual TMI concentrations which needs further evaluation and verification. In the revised MS, we deleted the reference of Tao et al. (2020). The influence of ionic strength on the reaction rates were shown in the SI Fig.S1 and discussed in Section 2.1 in the original MS.

4 It is important that the model has considered the activity coefficient values and reactions about oxalate and Fe. So how about the concentration of oxalate used in the model?

Due to the lack of direct measurements in the mentioned field observation campaign, we calculated the aerosol oxalate concentration according to Tao and Murphy (2019) which indicated a mechanisms responsible for the interactions among oxalate, pH, and Fe dissolution in PM2.5 based on a long term records in urban and rural areas. The linear regression between monthly average oxalate (nmol/m3air) and water-soluble Fe (nmol/m3air) concentration in PM2.5 was fitted as y=2.89x+0.27 with R2 as 0.68. Averaged oxalate aqueous concentration in winter field campaign were 0.55±0.42 in clean period, 0.82±0.48 in slightly polluted period, 0.38±0.29 in polluted period and 0.15±0.16 in highly polluted period with the Fe solubility as 5% in PM2.5. We also added the above paragraph in the revised SI Section Text S2.

**Other minor comments:**

5 Line 11, the wording of sulfate should be better consistent, "sulphate" or "sulfate". We changed all word "sulphate" to "sulfate" in the revised MS.

6 Line 13, the statement of "observed concentrations of transition metal ions" is not appropriate from my perspective, given the authors only measured the total concentration of Fe and Cu.

We deleted the words "ions (TMI)" in the revised MS abstract as "...using a state-ofart multiphase model constrained to the observed concentrations of transition metal, nitrogen dioxide, ozone, ..."

7 Line 22, "...affect the environmental quality and human health", references to support this conclusion are lacking.

We added the references in the revised MS as "Secondary sulfate aerosol is an important component of fine particles in severe haze periods (Zheng et al., 2015; Huang et al., 2014; Guo et al., 2014), which adversely affect the environmental quality and human health (Lippmann and Thurston, 1996; Fang et al., 2017; Shang et al., 2020)."

8 Line 149, "Obvious correlations between alpha-Fe (III) and sulfate...", The author may better calculate the  $R^2$  of alpha-Fe (III) and sulfate.

In four haze periods mentioned in the PKU-17 field campaign, the correlation coefficient  $R^2$  was 0.71, and in WD-14 field campaign haze periods, the coefficient was 0.63 indicating the obvious correlations between aqueous TMI and sulfate formation rates.

In section 2.2 in the revised MS:

"Obvious correlations between  $\alpha$ Fe (III) and sulfate concentration shown in **Fig. 1 (c)** and (d) were observed in the haze periods both in summer (R2=0.63) and winter (R2=0.71) and the correlation is consistent with the important contributions from aqTMI pathway to the sulfate formation."

9 Line 380, in Fig. 1(d), the modeled sulfate concentration line is missing.

We added the modeled sulfate concentration line in the Figure 1 panel (d). Because of the higher boundary layer height and more active lateral boundary conditions in summer, the simulations of secondary sulfate were not in line with observed sulfate concentration. In the section Method 4, we clarified the uncertainties of summer simulated sulfate concentration.

"Due to the higher and more dramatically diurnal changing BLH in summer (Lou et al., 2019), and the lack of relevant data in WD-14 field campaign, we could not get the modelled results of sulfate concentrations in summer haze periods. Direct emissions and transport of sulfate were not considered in the calculation because secondary sulfate is the predominant source in winter haze periods. Dilution was not considered either because the atmosphere is relatively homogeneous during winter haze episodes. Since haze events are normally accompanied by a low boundary layer height (Ht), Ht was set at 300 m at night-time and 450 m at noon (Xue et al., 2016). At other times, Ht was estimated using a polynomial (n = 2) regression as recommended in previous study (Xue et al., 2016). The diurnal trends of sulfate concentrations of the winter haze period using the deposition velocity of 1.5 cm/s and of 2 cm/s in summer are shown in Fig. 1 (c) and (d). Model results had the same trend with the observed values and could explain

the missing source of sulfate aerosol to some extent in winter while with high uncertainties in summer condition."